# Pask integrates hormonal signaling with histone modification via Wdr5 phosphorylation to drive myogenesis

Chintan K Kikani[1], Xiaoying Wu[1], Litty Paul[2,3], Hana Sabic[1], Zuolian Shen[4], Arvind Shakya[4], Alexandra Keefe[5], Claudio Villanueva[1], Gabrielle Kardon[5], Barbara Graves[2,3,6], Dean Tantin[4], Jared Rutter[1,6]*

[1]Department of Biochemistry, University of Utah School of Medicine, Salt Lake City, United States; [2]Department of Oncological Sciences, University of Utah School of Medicine, Salt Lake City, United States; [3]Huntsman Cancer Institute, University of Utah School of Medicine, Salt Lake City, United States; [4]Department of Pathology, University of Utah School of Medicine, Salt Lake City, United States; [5]Department of Human Genetics, University of Utah School of Medicine, Salt Lake City, United States; [6]Howard Hughes Medical Institute, University of Utah School of Medicine, Salt Lake City, United States

**Abstract** PAS domain containing protein kinase (Pask) is an evolutionarily conserved protein kinase implicated in energy homeostasis and metabolic regulation across eukaryotic species. We now describe an unexpected role of Pask in promoting the differentiation of myogenic progenitor cells, embryonic stem cells and adipogenic progenitor cells. This function of Pask is dependent upon its ability to phosphorylate Wdr5, a member of several protein complexes including those that catalyze histone H3 Lysine 4 trimethylation (H3K4me3) during transcriptional activation. Our findings suggest that, during myoblast differentiation, Pask stimulates the conversion of repressive H3K4me1 to activating H3K4me3 marks on the promoter of the differentiation gene myogenin (*Myog*) via Wdr5 phosphorylation. This enhances accessibility of the MyoD transcription factor and enables transcriptional activation of the *Myog* promoter to initiate muscle differentiation. Thus, as an upstream kinase of Wdr5, Pask integrates signaling cues with the transcriptional network to regulate the differentiation of progenitor cells.

*For correspondence: rutter@biochem.utah.edu

Competing interests: The authors declare that no competing interests exist.

## Introduction

Pask (PAS domain containing protein Kinase) is an evolutionarily conserved protein kinase that has been implicated in signaling to coordinate nutrient sensing with metabolic control across eukaryotic phylogeny (*Hao and Rutter, 2008*). The principal modality by which mammalian Pask appears to exert metabolic control is the regulation of gene expression. For example, Pask was shown to regulate the synthesis of fatty acids and triglycerides in the liver via activation of Sterol Regulatory Element Binding Protein-1 (SREBP-1) transcriptional activity in response to feeding and insulin (*Wu et al., 2014*). Probably related to this function, pharmacologic inhibition or genetic ablation of Pask resulted in decreased liver fat content and improved insulin sensitivity in rodent models of diabetes and obesity (*Hao et al., 2007*; *Wu et al., 2014*). In pancreatic β-cells, Pask responds to extracellular glucose and stimulates the transcription of the gene encoding insulin via regulation of the PDX-1 transcription factor (*An et al., 2006*; *Semache et al., 2013*). In spite of these observations, it has remained unclear how Pask coordinates these seemingly diverse transcriptional responses in different cell types.

Pask is expressed at a low level in most adult tissues (*Katschinski et al., 2003*). Interestingly, we noticed an elevated *Pask* mRNA abundance in stem or progenitor cell types in several transcriptome datasets. Using genetic and pharmacologic means of modulating Pask activity, we have uncovered a novel function of Pask in regulating the differentiation of stem and progenitor cells into neuronal, adipocytes or myocytes lineages. The mechanism underlying this role depends upon direct phosphorylation of Wdr5, which is a component of several chromatin modifying complexes, including mixed lineage leukemia (Mll) histone H3 Lysine 4 (H3K4) methyltransferase complexes (*Ruthenburg et al., 2007*; *Wysocka et al., 2005*). Wdr5 is a histone H3 binding protein (*Wysocka et al., 2005*) that is postulated to present the H3 N-terminal tail to the Mll or Set1 enzymes for methylation at lysine 4 (*Ruthenburg et al., 2006*; *Schuetz et al., 2006*).

Lysine 4 of Histone H3 is sequentially methylated to the mono- (H3K4me1), di- (H3K4me2) and tri-methyl (H3K4me3) forms by methyltransferases (*Shilatifard, 2012*). H3K4me1 is typically found at enhancers, which are binding sites for regulatory DNA-binding transcription factors (*Rada-Iglesias et al., 2011*; *Shlyueva et al., 2014*). However, a recent study demonstrated that H3K4me1 functions as a transcriptional repressive mark at the promoters of lineage specifying genes (*Cheng et al., 2014*). In contrast, H3K4me3 marks are usually associated with transcriptionally active promoters, or with poised promoters when found together with repressive H3K27me3 marks (*Bernstein et al., 2006*). These histone modifications collaborate with pioneering transcription factors to elicit programs of gene expression that drive differentiation of stem and progenitor cells (*Zaret and Carroll, 2011*). Using myogenic progenitor cells as a model of inducible differentiation, we show that phosphorylation of a single Wdr5 serine by Pask is necessary and sufficient for the conversion of repressive H3K4me1 marks to activating H3K4me3 marks at the lineage-specifying myogenin (*Myog*) promoter. This concomitantly enhances MyoD recruitment and chromatin remodeling of the *Myog* promoter and stimulates transcription of *Myog* to initiate terminal differentiation. Taken together, our results establish Wdr5 phosphorylation by Pask as an important node in the signaling and transcriptional network that initiates and executes differentiation.

## Results

### Pask is required for terminal differentiation in multiple cell lineages in vitro and muscle regeneration in vivo

As part of our ongoing study of the regulation and function of Pask, we examined *Pask* mRNA abundance in several publicly available gene expression datasets. We observed elevated *Pask* mRNA across diverse stem and progenitor cell types compared to differentiated cells and tissues (*Figure 1—figure supplement 1A*). For example, *Pask* was more abundant in mouse embryonic stem (ES) cells and progenitor cell types such as C2C12 myoblasts, C3H10T1/2 mesenchymal stem cells, Neuro2a neuroblastoma cells and immune progenitor cells compared to mouse embryonic fibroblasts, other somatic cell types and adult tissues (*Figure 1—figure supplement 1A*) (BioGPS:Pask, GeneAtlas MOE430). Furthermore, we noticed an increase in *Pask* expression during reprogramming of hepatocytes, fibroblasts and melanocytes to induced pluripotent stem cells (iPSCs). The increased *Pask* expression in iPSCs was comparable to the abundance observed in undifferentiated ES cells (*Figure 1—figure supplement 1B*) (*Ohi et al., 2011*). Conversely, terminal differentiation of human ESCs into cardiac muscle resulted in a progressive decline in the *Pask* expression before ultimately reaching the low abundance found in the adult heart (*Figure 1—figure supplement 1C*) (*Cao et al., 2008*) suggesting a positive correlation between *Pask* expression and stemness.

In examining potential drivers of *Pask* expression in transcription factor ChIP-Seq databases from mouse ESCs, we noticed that the *Pask* promoter was occupied by the Oct4 and Nanog pluripotency transcription factors (*Figure 1—figure supplement 1D*) (*Marson et al., 2008*). The Oct4 and Nanog binding region of the *Pask* promoter is evolutionarily conserved and is decorated in ESCs with transcriptionally favorable histone H3 lysine 27 acetylation (H3K27ac) and H3 Lysine 4 trimethylation (H3K4me3) modifications and RNA Polymerase II recruitment, suggestive of transcriptional activation (*Figure 1—figure supplement 1D*) (*Karlic et al., 2010*; *Moorefield, 2013*). Silencing of Oct4 or Nanog in ES cells resulted in modest, but statistically significant, suppression of *Pask* expression (*Figure 1—figure supplement 1E*) (*Loh et al., 2006*). Thus, gene expression and promoter analysis

suggest robust stem and progenitor cell-specific expression of *Pask*, possibly driven by the Oct4 and Nanog transcription factors, in embryonic stem cells.

To determine if this high *Pask* expression in stem cells, including in iPSCs, is indicative of Pask being functionally important for either iPSC generation or differentiation, we first assessed the effect of Pask inhibition on iPSC reprogramming from mouse embryonic fibroblasts (MEFs). Extensive studies using in vitro, cell and animal models have established the Pask selectivity and lack of toxicity of the BioE-1197 Pask inhibitor, making it a reliable and acute means of suppressing Pask activity in cells (*Wu et al., 2014*). In the presence or absence of BioE-1197, we induced reprogramming of mouse embryonic fibroblasts (MEFs) expressing IRES-driven Green Fluorescent Protein (GFP) from the endogenous *Pou5f1 (Oct4)* locus (*Lengner et al., 2007*), which enables robust GFP expression upon successful reprogramming. As shown in *Figure 1—figure supplement 2A–B*, Pask inhibition had no effect on the GFP+ colony formation, indicating that Pask is dispensable for iPSC reprogramming.

We next examined the necessity of Pask for differentiation of pluripotent ES cells into neurons using retinoic acid (*Kim et al., 2009*) in the presence or absence of BioE-1197. Differentiation in control cells caused Oct4 expression to be lost and increased expression of the neuron-specific Glycine Receptor Alpha 1 (Glra1) (*Figure 1—figure supplement 2C*). In contrast, each of these signs of terminal differentiation was lost or blunted by treatment with BioE-1197 (*Figure 1—figure supplement 1H*), suggesting a necessity of Pask for neuronal differentiation of ES cells.

Progenitor cells such as C3H10T1/2 mesenchymal stem cells and C2C12 myoblasts also robustly express *Pask* (*Figure 1—figure supplement 3A*). C3H10T1/2 cells differentiate into adipocytes in response to appropriate signaling cues (*Pinney and Emerson, 1989*; *Reznikoff et al., 1973*; *Zhou et al., 2013*). To determine if Pask is required for the differentiation of these progenitor cells similar to neuronal differentiation of ES cells, we treated C3H10T1/2 cells with adipogenic stimuli in the presence or absence of BioE-1197. Vehicle-treated control cells efficiently differentiated into mature adipocytes, as evidenced by the accumulation of lipid droplets in more than 80% of cells (*Figure 1—figure supplement 3A*, quantified in *Figure 1—figure supplement 3B*). In contrast, Pask inhibition greatly impaired adipocyte differentiation, as only about 10% of cells contained observable lipid droplets. We also observed impaired expression of the *Adipoq* and *Fabp4* adipocyte markers (*Zhou et al., 2013*) (*Figure 1—figure supplement 3C*) in BioE-1197 treated samples compared with DMSO control. BioE-1197 had no effect on cell doubling time in proliferative C3H10T1/2 cells (*Figure 1—figure supplement 3D*).

We also evaluated the necessity of Pask for differentiation of mouse C2C12 muscle progenitor cells, which differentiate into multi-nucleated muscle fibers in response to horse serum or insulin (*Figure 1A*, control) (*Yaffe and Saxel, 1977*). Pooled siRNA knockdown or inhibition of Pask strongly impaired the formation of multi-nucleated myotubes (*Figure 1A*, quantified in *Figure 1B*, *Figure 1—source data 1*). Similarly, CRISPR/Cas9-mediated *Pask* mutation also resulted in suppression of insulin-induced differentiation as indicated by the absence of multi-nucleated myotubes (*Figure 1—figure supplement 4A*, quantified in 4B) and prevented the induction of myosin heavy chain (MHC) protein expression (*Figure 1—figure supplement 4C*). In addition, BioE-1197 treatment also blunted the induction of myosin heavy chain in human primary myoblasts in response to both horse serum and insulin treatment (*Figure 1—figure supplement 4D*). Similar to mouse myoblasts, induction of human *MYLPF* (myosin light chain) and *ACTA1* (skeletal muscle actin) mRNAs was also blunted or abrogated by Pask inhibition during differentiation of human myoblasts (*Figure 1—figure supplement 4E*).

The defect in myotube formation caused by knockdown of endogenous mouse *Pask* was reversed by the expression of siRNA-resistant human wild type (WT) Pask, but not a K1028R kinase-dead (KD) Pask mutant (*Figure 1C*, quantified in *Figure 1D*, *Figure 1—source data 1*) (*Kikani et al., 2010*), indicating the necessity of Pask catalytic activity for myoblast fusion. Importantly, this effect appears to be cell autonomous as only that fraction of cells in the population that express WT Pask (WT+) show rescue of the fusion defect, whereas cells not expressing hPask within the same culture (WT-) remained mono-nucleated and fusion defective. Additionally, *Mylpf* (myosin light chain) and *Acta1* mRNAs, which are both markers of myoblast differentiation, failed to be induced in *Pask* knockdown cells. Expression of both genes was rescued by expression of WT but not KD human PASK (*Figure 1E*).

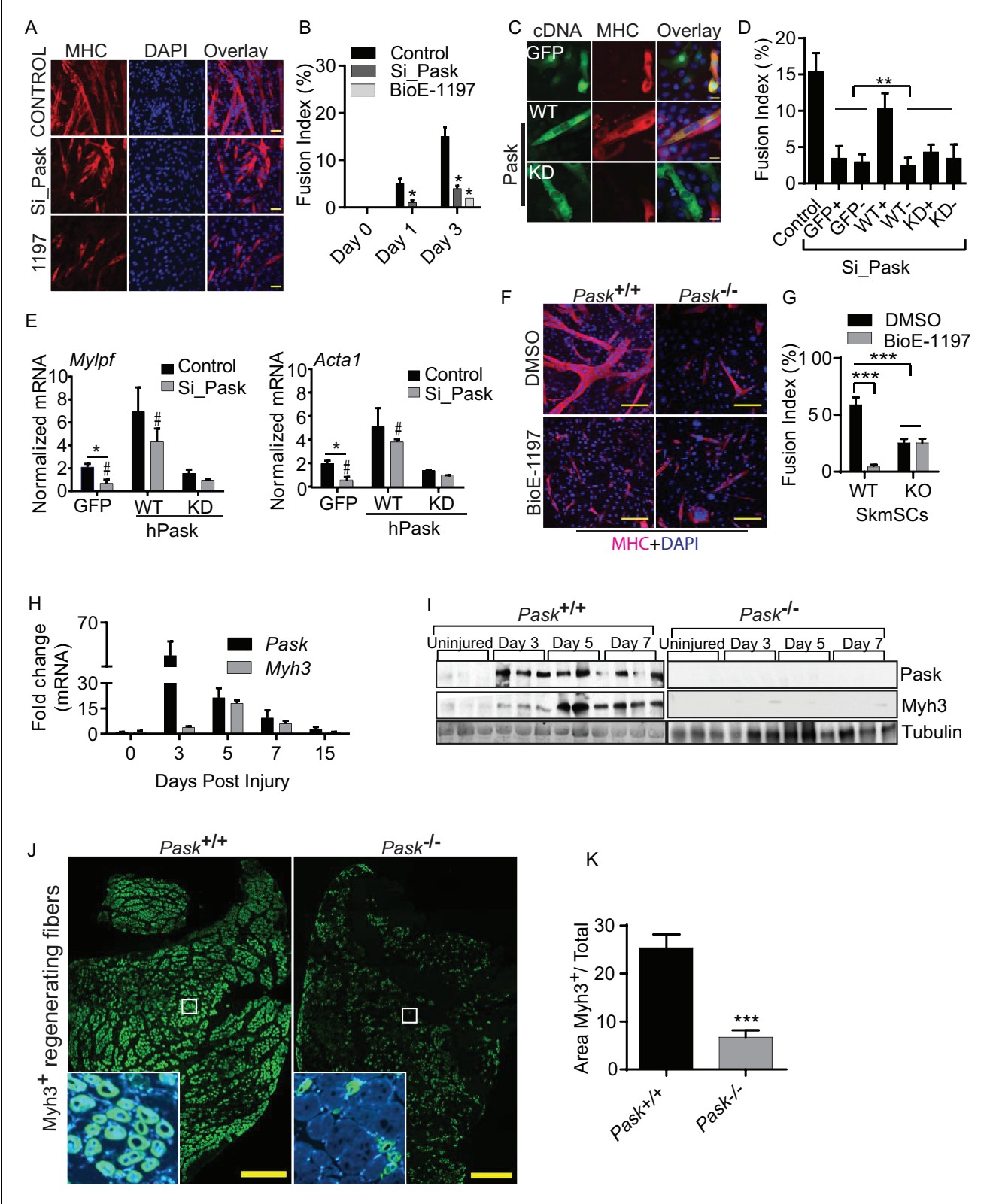

**Figure 1.** Pask is required for skeletal muscle regeneration after acute muscle injury. (**A**) Differentiation of C2C12 myoblasts was assessed after Pask was either knocked down using pooled siRNA or inhibited using 25 μM BioE-1197. Control cells were transfected with pooled non-targeting siRNA. DMSO vehicle control (v/v) was used for BioE-1197 samples and it was indistinguishable from siRNA control. 48 hr after siRNA or BioE-1197 treatment, differentiation was stimulated using 10nM insulin and myotube formation was visualized using anti-MHC (MF20-Red) antibody 3 days later. Scale bar =

*Figure 1 continued on next page*

*Figure 1 continued*

40 µM. (**B**) Comparison of fusion index between control and siRNA (Si_Pask) or inhibitor (BioE-1197)-treated C2C12 cells from (**A**). n = 3 independent experiments each with 100 cells counted. Error bars ± S.D. *p<0.05 (**C**) GFP, Flag-WT or Flag-KD (K1028R) human Pask were expressed using retrovirus in *Pask*-silenced C2C12 cells. Differentiation was initiated 24 hr after transgene introduction and was analyzed using anti-MHC staining on day 3 as in (**A**). Expression of Pask was visualized using anti-Flag staining. Scale bar = 20 µm. (**D**) Fusion index (as in (**B**)) of cells from (**C**) that were individually scored as with (+) or without (−) GFP, WT Pask or KD Pask expression. Note that only those cells expressing WT Pask show restoration of fusion index. n = 3 independent experiments each with 100 cells counted. Error bars ± S.D. **p<0.005. (**E**) qRT-PCR analysis of C2C12 cells showing abundance of *Mylpf* and *Acta1* mRNAs in *Pask*-silenced cells expressing GFP, WT or KD Pask. 18S rRNA was used as normalizer. Error bars ± S.D. *p<0.05, #p<0.05 WT vs KD Pask in control samples, #p<0.05 WT vs KD hPask in Si_Pask samples. (**F**) Myogenesis of primary myoblasts derived from WT and *Pask*$^{-/-}$ skeletal muscle was assessed using anti-MHC staining after four days of differentiation. Nuclei are stained with DAPI. Scale bar = 100 µM. (**G**) Quantification of fusion index from (**F**) at Day 4. ***p<0.0005. (**H**) qRT-PCR analysis of fold change in the expression of *Pask* and *Myh3* mRNA following BaCl$_2$ induced muscle injury to TA muscle relative to uninjured (DPI 0). 18S rRNA was used as normalizer. (**I**) Western blot analysis of isolated TA muscle following BaCl$_2$ induced muscle injury from WT and *Pask*$^{-/-}$ mice. (**J**) Representative cross-section of TA muscle 5d post injury showing levels of Myh3 (green) expression between WT (Pask$^{+/+}$) and Pask$^{-/-}$ animals. Nuclei are stained with DAPI (blue). Scale bars = 100 µM. n = 5 (**K**) Quantification of the Myh3 positive area/total was determined by measuring Myh3 staining intensities across 3 representative sections from each independent animal, n = 5. Error bars ± S.D. ***p<0.0005.

The following source data and figure supplements are available for figure 1:

**Source data 1.** Numerical values from graphs representted in *Figure 1*.
**Figure supplement 1.** Pask is enriched in stem cells and is required for terminal differentiation of neuronal, adipogenic and myogenic cell types.
**Figure supplement 2.** Pask is required for differentiation of ES cells but not for iPS reprograming.
**Figure supplement 3.** Pask is required for adipogenesis of mesenchymal stem cell.
**Figure supplement 4.** Genetic and pharmacological inhibition of Pask suppresses myogenesis of mouse and human myoblasts.

We also examined the proliferation and differentiation of primary myoblasts isolated from hindlimb muscles of WT and *Pask*$^{-/-}$ mice in the presence or absence of BioE-1197. Under proliferative culture conditions, neither genetic deletion nor inhibition of Pask had any effect on the proliferation rate (*Figure 1—figure supplement 4F*, *Figure 1—source data 1*). In response to differentiation cues, myoblasts derived from WT mice efficiently formed multi-nucleated myotubes (*Figure 1F*, quantified in *Figure 1G*). In contrast, genetic elimination of Pask (i.e. myoblasts derived from germline *Pask*$^{-/-}$ mice) caused a ~60% decrease in differentiation efficiency. Interestingly, the effect of Pask inhibition in WT cells was much more pronounced, almost completely eliminating myotube formation. BioE-1197 had no effect, however, on *Pask*$^{-/-}$ myoblasts (*Figure 1F*, quantified in *Figure 1G*). This result demonstrates that the effects of the BioE-1197 Pask inhibitor are completely dependent on the presence of Pask, a clear indication that these effects are mediated by Pask inhibition. Moreover, these results demonstrate that myoblasts derived from germline *Pask*$^{-/-}$ mice exhibit a Pask-independent compensation that appears, at least in vitro, to partially rescue myotube formation.

The myoblast differentiation process is activated in vivo to repair and replenish the myotome after muscle injury. During muscle regeneration, satellite cells become activated, proliferate and eventually fuse in response to differentiating signals to restore mature myotubes. We observed that as early as 3 days post injury, PASK mRNA (*Figure 1H*, *Figure 1—source data 1*) and protein (*Figure 1I*) expression was robustly upregulated. This is consistent with high Pask expression in stem cells compared with differentiated cells (*Figure 1—figure supplement 1A*), since muscle stem cells begin to proliferate upon muscle injury before eventually fusing to replenish the injured myotome. To test the functional relevance of increased *Pask* expression during muscle regeneration, we compared the kinetics of muscle regeneration in WT and *Pask*$^{-/-}$ mice. As shown in *Figure 1I*, whereas Myh3, a marker of regenerative myogenesis, was robustly induced as early as Day 5 in WT animals, *Pask*$^{-/-}$ animals showed very little induction of Myh3 expression. Similarly, WT animals show robust induction of Myh3 immunohistochemical staining, but this was blunted in *Pask*$^{-/-}$ muscle (*Figure 1J*, quantified in *Figure 1K*). Taken together, our data suggest that mammalian Pask plays an important

role in the differentiation of stem cells across many lineages in vitro and during muscle regeneration in vivo.

## Pask and Myogenin establish a positive transcriptional feedback loop that enforces myocyte differentiation

Following injury, satellite cells execute a transcriptional program that culminates in the formation of syncytial myocytes that constitute a healthy muscle fiber (*Figure 2A*) (*Bentzinger et al., 2012*). Quiescent Pair Box 7[+] (Pax7[+]) satellite cells are induced to proliferate and initiate expression of Myoblast Determination protein (MyoD) and/orMyogenic Factor 5(Myf5). In MyoD[+] myoblasts, MyoD drives the expression of Myogenin (MyoG) in response to differentiation cues. MyoG then executes the terminal differentiation program by down-regulating Pax7 expression (*Olguin and Olwin, 2004*) and activating the genes necessary for myoblast fusion and muscle function, including myosin heavy chain (*Mylpf*) and muscle specific actin (*Acta1*).

To identify the underlying mechanism for the defect in myoblast differentiation caused by loss of Pask activity, we compared the expression of these key transcription factors and their targets during the differentiation of WT and *Pask*[-/-] satellite cells. As expected, the abundance of the *Pax7* and *Myf5* mRNAs declined at day 1 of differentiation in WT cells (*Figure 2B*, *Figure 2—source data 1*) (*Bentzinger et al., 2012*). In conjunction, *Myog* mRNA abundance increased at Day 1, as did expression of its target genes *Mylpf* and *Acta1* (*Figure 2B*, *Figure 2—source data 1*). *Pask*[-/-] cells exhibited, even at Day 0, a more satellite cell-like gene expression profile, with *Pax7* and *Myf5* being elevated relative to WT and *Myog* and its targets *Mylpf* and *Acta1* having decreased expression. At Day 1 of differentiation, these patterns were maintained, with *Pax7* and *Myf5* mRNAs remaining elevated and *Myog*, *Mylpf* and *Acta1* remaining decreased (*Figure 2B*, *Figure 2—source data 1*). The expression of *Myod*, on the other hand, was not affected by the loss of Pask. We also compared the protein levels of Pask, Pax7 and MyoG during in vivo muscle regeneration between WT and *Pask*[-/-] animals (*Figure 2C*). Consistent with the mRNA expression data from satellite cells, Pax7 protein was aberrantly elevated in *Pask*[-/-] animals compared with WT animals at Day 5 post-injury. *Pask*[-/-] animals also showed a profound defect in the induction of MyoG and its target Myh3 (*Figure 2C*).

To determine if these effects of genetic loss of Pask can be recapitulated by acute means of controlling Pask expression or activity in C2C12 myoblasts, we examined mRNA or protein levels of Pax7, MyoG and MHC after Pask knockdown or inhibition (*Figure 2—figure supplement 1A–B*). At Day 1 of differentiation, when Pask silencing was most effective, Pax7 abundance remained elevated and the induction of the early differentiation marker MyoG and its target MHC was delayed in Pask-silenced C2C12 cells (*Figure 2—figure supplement 1A*). Pask inhibition by BioE-1197 recapitulated these effects at the mRNA level during insulin-induced C2C12 differentiation (*Figure 2—figure supplement 1B*). These transcriptional defects in myogenesis caused by *Pask* knockdown were rescued by ectopic expression of human WT but not kinase dead (KD) Pask (*Figure 2—figure supplement 1C*). Taken together, our findings suggest that Pask is required to initiate the commitment phase of myoblast differentiation and maintain the myogenic transcriptional program.

To properly place Pask in the myogenic transcription factor cascade during differentiation (*Figure 2A*), we profiled by co-immunostaining the expression of endogenous Pask with Pax7, MyoD or MyoG in proliferating (Day 0) and differentiating (Day 1 and 3) myoblasts in control, BioE-1197 treated or *Pask* knockdown conditions. At Day 0, ~40% of control cells were Pax7[+] and this fraction declined over the course of differentiation (*Figure 2D*, quantified in 2E, *Figure 2—source data 1*). When Pask was inhibited or knocked down, the number of Pax7[+] cells was elevated at Day 0 and during differentiation, matching the mRNA data (*Figure 2B*, *Figure 2—figure supplement 1A–B*). In contrast, *Pask* knockdown or inhibition had no effect on the number or subcellular localization of MyoD at any point during differentiation, consistent with *Figure 2B* (*Figure 2—figure supplement 2A*, quantified in *Figure 2—figure supplement 2B*). The effects of loss of Pask activity were most obvious in the appearance of MyoG[+] cells during differentiation. In control cells, the fraction of MyoG[+] cells dramatically increased from Day 0 (~7%) to Day 1 (~60%) and Day 2 (~80%) (*Figure 2F*, quantified in 2G, *Figure 2—source data 1*). MyoG[+] cell numbers were substantially reduced by either Pask inhibition or knockdown. We also noticed a striking correlation of Pask and MyoG expression. At Day 1, we observed a strong co-expression between Pask and MyoG, wherein ~50% of Pask[+] cells were MyoG[+] whereas only ~15% of Pask[-] cells were MyoG[+] (*Figure 2F*, quantified in *Figure 2H*). As expected, inhibition of Pask function with BioE-1197 eliminated this correlation, with

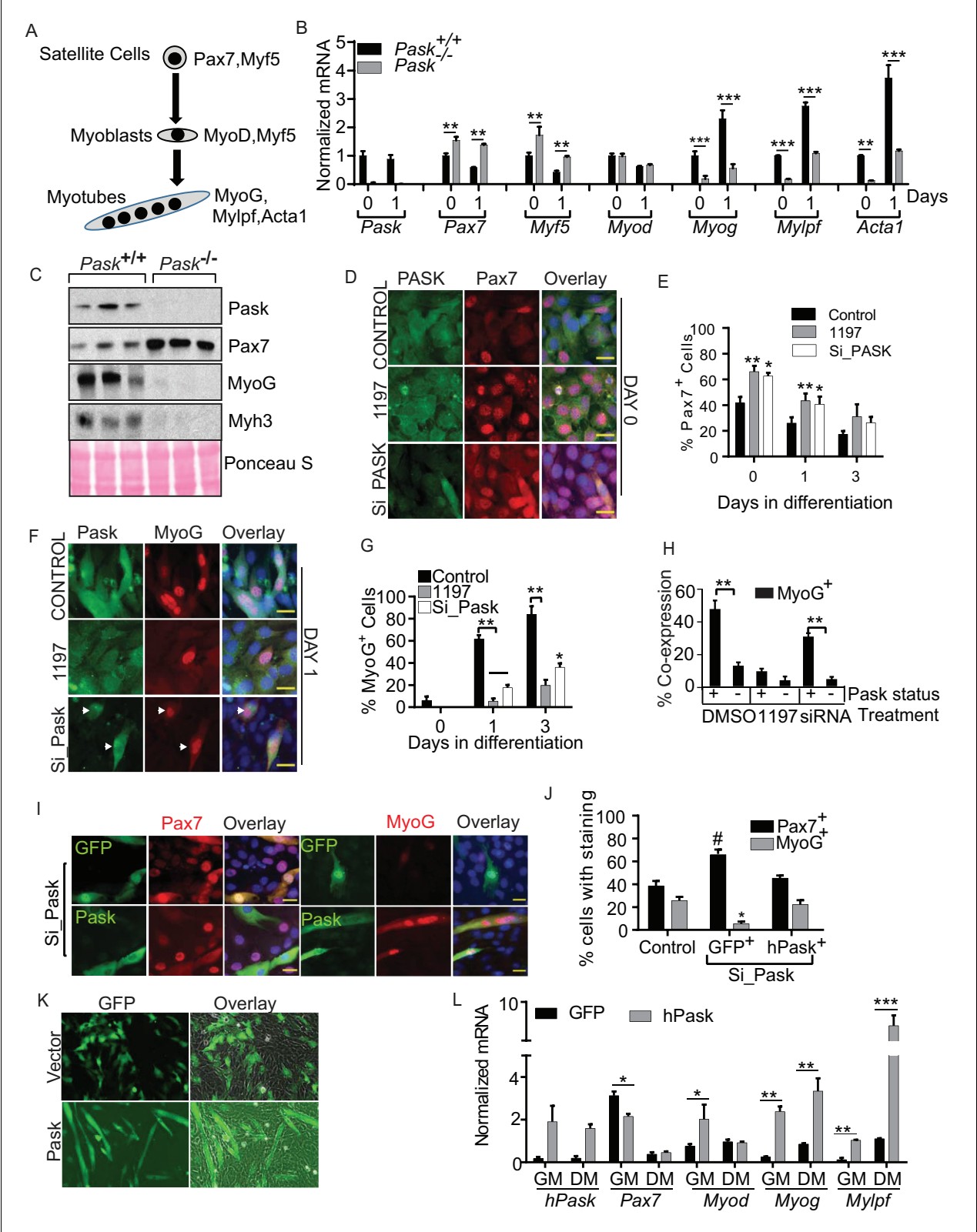

**Figure 2.** Pask is required for transcriptional activation of MyoG in response to differentiation cues. (**A**) Schematic of myogenesis from satellite cells that depicts the progression of transcription factors during myogenesis. (**B**) qRT-PCR analysis of WT and *Pask*<sup>-/-</sup> satellite cells prior to (day 0) or at the indicated time after initiation of differentiation with 100 nM insulin in serum free DMEM. 18S rRNA was used as normalizer. Transcript levels of WT cells at Day 0 was set at 1 to calculate fold changes during differentiation. Error bars ± S.D. *p<0.05, **p<0.005, ***p<0.001. (**C**) Western blot analysis of the
*Figure 2 continued on next page*

*Figure 2 continued*

indicated proteins at five days post-injury from isolated TA muscles of WT and *Pask*<sup>-/-</sup> mice. (D) Immunofluorescence microscopic examination of Pax7 expression in control, Pask-siRNA or 25μM BioE-1197 treated samples on Day 0 of differentiation. (E) Quantification of Pax7+ cell numbers from experiment in (D) along the differentiation time course. n=3 independent experiments each with 100 cells counted. Error bars = S.D. *p<0.05. (F) Immunofluorescence microscopy showing MyoG and Pask in control, Pask-siRNA or 25 μM BioE-119- treated cells at Day 1 of differentiation. (G) Percent MyoG⁺ cells in Control, *Pask*-siRNA or 25 μM BioE-1197-treated cells during differentiation as in (F). n = 3 independent experiments each with 100 cells counted. Error bars ± S.D. **p<0.005, *p<0.05. (H) Percent co-expression of Pask⁺ and MyoG⁺ cells at Day 1 of differentiation as in (F). n = 3 independent experiments each with 100 cells counted. Error bars ± S.D. **p<0.005. (I) Representative images from *Pask*-silenced C2C12 cells expressing GFP control or Flag-human Pask at 24 hr after initiation of differentiation to detect endogenous Pax7 (left) or MyoG (right) together with GFP or Flag-Pask. Scale bar = 20 μM. (J) Quantification of Pax7⁺ and MyoG⁺ cells counting only those cells that are GFP⁺ or Pask⁺ from (I). n = 3 independent experiments each with 100 cells counted. Error bars ± S.D. *p<0.05 Control vs. Pask silenced GFP⁺ cells, #p<0.05 Pask silenced GFP⁺ vs. Pask silenced hPask⁺ cells. (K) Empty GFP vector or GFP vector containing hPask were retrovirally introduced to proliferating C2C12 cells at a sub-confluent density in growth media. Cells were allowed to grow in growth media for 72 hr and imaged for GFP. (L) qRT-PCR analysis of GFP or hPask-expressing C2C12 myoblasts after puromycin selection from (I). Cells were collected from growth media (GM) or 24 hr after addition of differentiation media (DM). n = 3, Error bars ± S.D. *p<0.05, **p<0.005, ***p<0.0005.

The following source data and figure supplements are available for figure 2:

**Source data 1.** Numerical data from qPCR analysis off WT vs Pask-/- satellite cells from the graph represented in *Figure 2B* and quantification of Pax7+ and MyoG+ cell numbers.

**Figure supplement 1.** Pask is required for transcriptional activation of MyoG in response to differentiation cues.

**Figure supplement 2.** Pask does not regulate MyoD⁺ cell population.

**Figure supplement 3.** Pask promoter is occupied by MyoG and MyoD during differentiation.

both populations losing MyoG staining. Importantly, in the *Pask* knockdown samples, MyoG expression was present almost exclusively in those cells in the population that escaped *Pask* silencing (*Figure 2F*, quantified in *Figure 2H*).

Both the increase in Pax7⁺ cells and the decrease in MyoG⁺ cells observed upon *Pask* knockdown were completely rescued by expression of WT hPask (*Figure 2I*, quantified in *Figure 2J*). Surprisingly, transient over-expression of human Pask, even in the absence of extrinsic differentiation cues, was sufficient to induce formation of multi-nucleated myocytes (*Figure 2K*) and to induce the entire differentiation transcriptional program in growth medium (GM) (*Figure 2L*). This included suppression of *Pax7* as well as induction of *Myod*, *Myog* and *Mylpf*. Addition of differentiation medium (DM) further enhanced myoblast differentiation in hPask-expressing cells, as evidenced by the elevated expression of *Myog* and *Mylpf* (*Figure 2L*). Together, these data establish Pask as necessary for the timely execution of the terminal differentiation program in myoblasts and as sufficient to initiate that program even in the absence of extrinsic differentiation signals.

The robust cell autonomous correlation between Pask and MyoG expression is probably a reflection of the necessity of Pask for MyoG expression, but it also prompted us to explore the possibility that MyoG might also regulate Pask expression during differentiation. Interestingly, previous ChIP-Seq data had suggested that MyoG, and to a lesser extent MyoD, occupies the *Pask* promoter during C2C12 differentiation coincident with H3K4me3 modification and RNA-PolII recruitment (*Figure 2—figure supplement 3A*) (*Yue et al., 2014*). We independently validated that MyoG was indeed recruited to the proximal *Pask* promoter upon initiation of differentiation (*Figure 2—figure supplement 3B*) and that MyoG expression stimulates expression from a *Pask* promoter-driven reporter gene (*Figure 2—figure supplement 3C*). These data suggest the possibility that *Pask* expression in differentiating myoblasts might be driven by MyoG to further support commitment to differentiate. Thus, taken together, our data show that Pask and MyoG engage in a positive, self-reinforcing feedback loop to enforce the terminal differentiation program once it has been signaled to initiate.

## Pask collaborates with MyoD to drive MyoG expression and myogenesis

The gene expression data presented above suggest the possibility that either failure to suppress Pax7 expression or failure to induce MyoG expression underlies the differentiation defect caused by the absence of active Pask. Because Pask protein expression correlated most strongly with that of MyoG and MyoG is known to repress Pax7 expression (*Olguin et al., 2007*), we hypothesized that Pask collaborates with MyoD to induce MyoG expression. MyoD is a pioneering transcription factor that is sufficient to induce a myogenic cell fate even in non-muscle progenitor cells. C3H10T1/2 mesenchymal stem cells normally express the Peroxisome proliferator-receptor activated gamma 2 (PPARγ2) adipogenic transcription factor and efficiently differentiate into adipocytes in response to adipogenic differentiation cues (*Figure 3A*, ) (*Zhao et al., 2013*). The myogenic transcriptional program is epigenetically silenced in C3H10T1/2 cells, but it can be activated in response to MyoD expression (*Penn et al., 2004*; *Tapscott et al., 1988*). MyoD stimulates MyoG expression, which then collaborates with MyoD to establish myogenic commitment and repress *Pparγ2* expression (*Figure 3A*).

Using this experimental paradigm, we asked whether Pask is necessary for MyoD-dependent MyoG expression and myogenesis in C3H10T1/2 cells. As shown in *Figure 3B*, MyoD-induced expression of *Myog* and *Mylpf* was impaired by Pask inhibition while *MyoD* expression was unaffected. Moreover, *Pparγ2* expression, which was lost in control cells due to myogenic lineage conversion, was maintained in cells treated with BioE-1197 (*Figure 3B*, *Figure 3—source data 1*). In addition, as assessed by immunofluorescence microscopy, the fraction of MyoG$^+$ cells at either Day 1 or Day 2 of differentiation was markedly decreased by Pask inhibition, while MyoDpositivity was unaffected (*Figure 3C*, quantified in *Figure 3D*; *Figure 3—figure supplement 1A–B*, *Figure 3—source data 1*). In addition to the loss of MyoG expression, cells treated with BioE-1197 exhibited a profound failure in the formation of multi-nucleated myotubes (*Figure 3E*). Based on these results, we hypothesized that the differentiation failure of Pask-inhibited cells was not due to failed inactivation of Pax7, which is not expressed in C3H10T1/2 cells, but instead is due to an inability of MyoD to induce MyoG expression in the absence of Pask. Indeed, we found that ectopic expression of rat MyoG was sufficient to bypass the Pask requirement for *Mylpf* expression even though endogenous mouse *Myog* remained suppressed in BioE-1197 treated cells (*Figure 3F*). These data place Pask within the myogenic transcriptional cascade, with Pask being necessary to act in concert with MyoD to induce MyoG expression and myogenic differentiation.

## Pask phosphorylates Wdr5

Catalytically inactive Pask was unable to rescue the differentiation defect caused by silencing of endogenous Pask (see *Figure 1C–E*, *Figure 2—figure supplement 2C*) and a catalytic inhibitor mimicked genetic inactivation of Pask (*Figures 1A*, *2F–G*). We concluded, therefore, that Pask-dependent phosphorylation of one or more specific substrates must be required for the myogenic gene expression and differentiation. Since none of the known Pask substrates are likely to serve this role, we sought to identify new substrates that might mediate the effects of Pask on differentiation. We employed a strategy that combines experimental and bioinformatics approaches. Specifically, we identified all proteins that had been described to physically interact with Pask in high-throughput protein-protein interactome datasets (BIOGRID: Pask) and were identified in our own unpublished Pask interactome studies. We then filtered this dataset for proteins that contain the consensus phosphorylation motif for Pask ([HKR]-X-[KR]-X-X-[ST]) that we previously identified (*Kikani et al., 2010*). Wdr5 was the single candidate substrate protein that emerged, based on its interaction with Pask in our dataset and in a global affinity capture screen (*Ewing et al., 2007*).

We first sought to validate the physical interaction of Pask and Wdr5 and did so in C2C12 cells across a differentiation time-course. As shown in *Figure 4A*, Wdr5 associated with Pask during the proliferative phase (-1 and 0 days). However, that interaction was significantly enhanced following the onset of differentiation (Day 1) (*Figure 4A*). We also found that co-expressed epitope-tagged forms of Pask and Wdr5 co-immunoprecipitated and that this did not require Pask catalytic activity as the K1028R Pask mutant (KD) precipitated Wdr5 equivalently to WT Pask (*Figure 4B*).

Wdr5 is a member of several protein complexes that catalyze histone methylation or acetylation (*Migliori et al., 2012*; *Shilatifard, 2012*; *Trievel and Shilatifard, 2009*). To properly place the Pask-

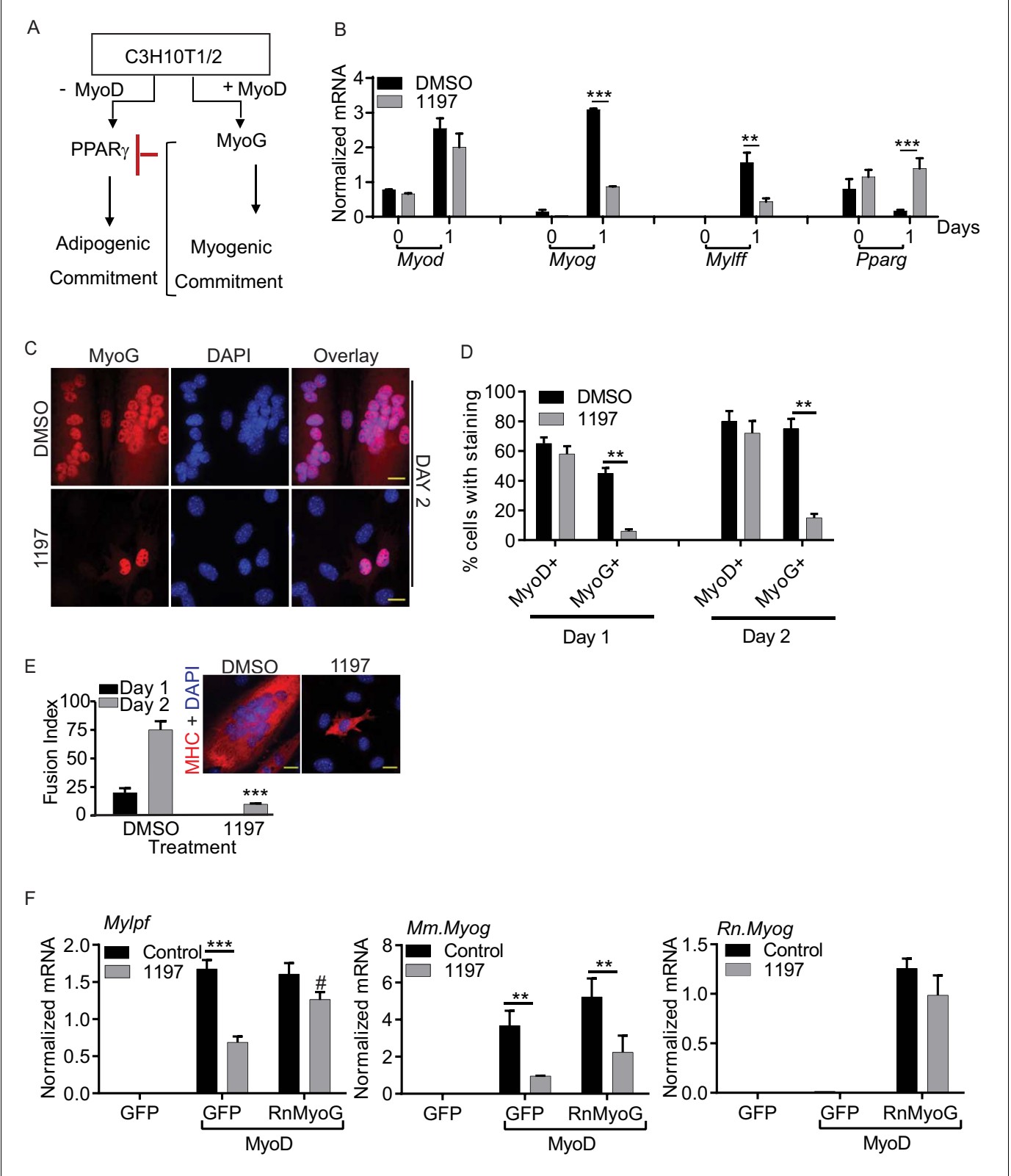

**Figure 3.** Pask is required for myogenic conversion of C3H10T1/2 cells by MyoD. (**A**) Schematic depiction of the mechanism by which MyoD-induces the myogenic conversion of adipogenic C3H10T1/2 cells. (**B**) qRT-PCR analysis of the indicated mRNAs in C3H10T1/2 cells expressing MyoD in the presence of DMSO or 25 μM BioE-1197 during differentiation. n = 3, Error bars ± S.D ***p<0.0005, p<0.005. (**C**) MyoD-expressing C3H10T1/2 cells were allowed to differentiate in the presence of DMSO or 25 μM BioE-1197 and processed for immunofluorescence microscopy using anti-MyoG antibody.
*Figure 3 continued on next page*

Figure 3 continued

Scale bar = 20 μm. (D) Quantification of MyoD[+] and MyoG[+] cells in the presence of DMSO or BioE-1197 from (C). n = 3 independent experiments each with 100 cells counted. Error bars ± S.D. **p<0.005. (E) Fusion index on Day 1 and 2 of differentiation for C3H10T1/2 cells expressing MyoD in the presence of DMSO or 25 μM BioE-1197. Inset shows representative MHC staining on Day 2 of differentiation. Scale bar = 20 μm. (F) qRT-PCR analysis of endogenous mouse *Mylpf* and *Myog* and rat *Myog* upon GFP or rat MyoG expression in MyoD-expressing C3H10T1/2 cells treated with either DMSO or 25 μM BioE-1197. n = 3, Error bars ± S.D. ***p<0.0005, **p<0.005.

The following source data and figure supplement are available for figure 3:

**Source data 1.** Numerical values from qPCR analysis in *Figure 3A* and quantification of MyoD+ and MyoG+ cell numbers from *Figure 3D*.

**Figure supplement 1.** Pask is required for myogenic conversion of mesenchymal stem cells by MyoD.

Wdr5 interaction within these complexes and to determine if Pask associates with any intact Wdr5-containing complexes, we expressed exclusive members of each of the major Wdr5- containing complexes. This included Menin (Mll1/2 complex), PTIP (Mll3/4 complex), GCN5 (Kat2A Histone Acetyl-transferase (HAT) complex) and Set9 (Setd7a – H3K4me3 complex) as well as Rbbp5, which is a member of the core Wdr5 sub-complex that is common to all Mll and Set complexes. Among these, only Wdr5 was able to co-purify endogenous Pask (*Figure 4—figure supplement 1A*), suggesting that Pask is not a stable member of intact Wdr5-containing complexes, but might associate selectively with free Wdr5. Supporting this notion, the interaction between Pask and Wdr5 appears to be direct as GST-tagged Wdr5, purified from bacteria, bound to Pask purified from insect cells (*Figure 4—figure supplement 1B*). Furthermore, while overexpressed Pask was cytoplasmic (typically perinuclear) in the majority of cells (*Rutter et al., 2001*), with only ~10% of cells exhibiting nuclear Pask in absence of Wdr5 co-expression, Wdr5 co-expression with Pask significantly enhanced the nuclear accumulation of Pask (*Figure 4C*). To define the minimal region of Pask that is responsible for Wdr5 interaction, we generated a series of N- and C-terminal Pask truncation mutations and assessed their ability to co-purify Wdr5. We found that the N-terminal 914 residues of Pask were dispensable for Wdr5 binding (*Figure 4—figure supplement 1C*). The interaction was lost, however, upon deletion of residues 915–948, suggesting that Wdr5 likely interacts with Pask just upstream of the canonical protein kinase domain.

Wdr5 was phosphorylated efficiently by WT Pask, but not KD Pask, in in vitro kinase reactions (*Figure 4D*) and this was abrogated by the BioE-1197 Pask inhibitor (*Figure 4—figure supplement 1D*). BioE-1197 also robustly blunted the in situ phosphorylation of Pask and Pask-bound Wdr5 using in-cell $^{32}$P labeling (*Figure 4E*). As Pask association with Wdr5 was enhanced at the onset of differentiation, we asked if Pask activity towards Wdr5 is stimulated at this time. Consistent with our previous report (*Kikani et al., 2010*), Pask exhibits modest activity in the absence of stimulation in C2C12 myoblasts (*Figure 4F*). At 12 hr post-addition of insulin containing differentiation media, however, Pask was activated as assessed by increased autophosphorylation and Wdr5 phosphorylation, thus suggesting that Pask kinase activity is stimulated during differentiation.

Wdr5 was selected based on it containing a sequence that matched the Pask consensus substrate motif (*Kikani et al., 2010*). That site, with Serine 49 as the putative phospho-acceptor residue, is strikingly similar to that of the best-characterized substrate of yeast Pask, Ugp1 (*Figure 4G*), which is also robustly phosphorylated by human Pask (*Kikani et al., 2010*; *Rutter et al., 2002*). In particular, the -5His and -3Lys residues, which were shown to be the most important for determining Pask phosphorylation (*Kikani et al., 2010*), are present in this putative Wdr5 phosphorylation motif. As expected, we found that Ser49 is required for phosphorylation by Pask since mutation to either glutamate (Glu) or alanine (Ala) nearly abolished phosphorylation in vitro (*Figure 4H*). Pask-associated WT Wdr5 was phosphorylated in cells as shown above, but the S49A mutant showed a marked reduction in $^{32}$P incorporation (*Figure 4I*). Together, these results suggest that Pask interacts with Wdr5 and phosphorylates it at Ser49 both in vitro and in cells.

Finally, we asked whether the phosphorylation status at Ser49 affects Wdr5 association with either Pask or members of its histone modifying complexes using unphosphorylatable S49A or phospho-mimetic S49E variants of Wdr5. Pask purified from insect cells bound to bacterial-expressed Wdr5^S49E more weakly than WT, whereas Wdr5^S49A bound Pask more avidly (*Figure 4—figure*

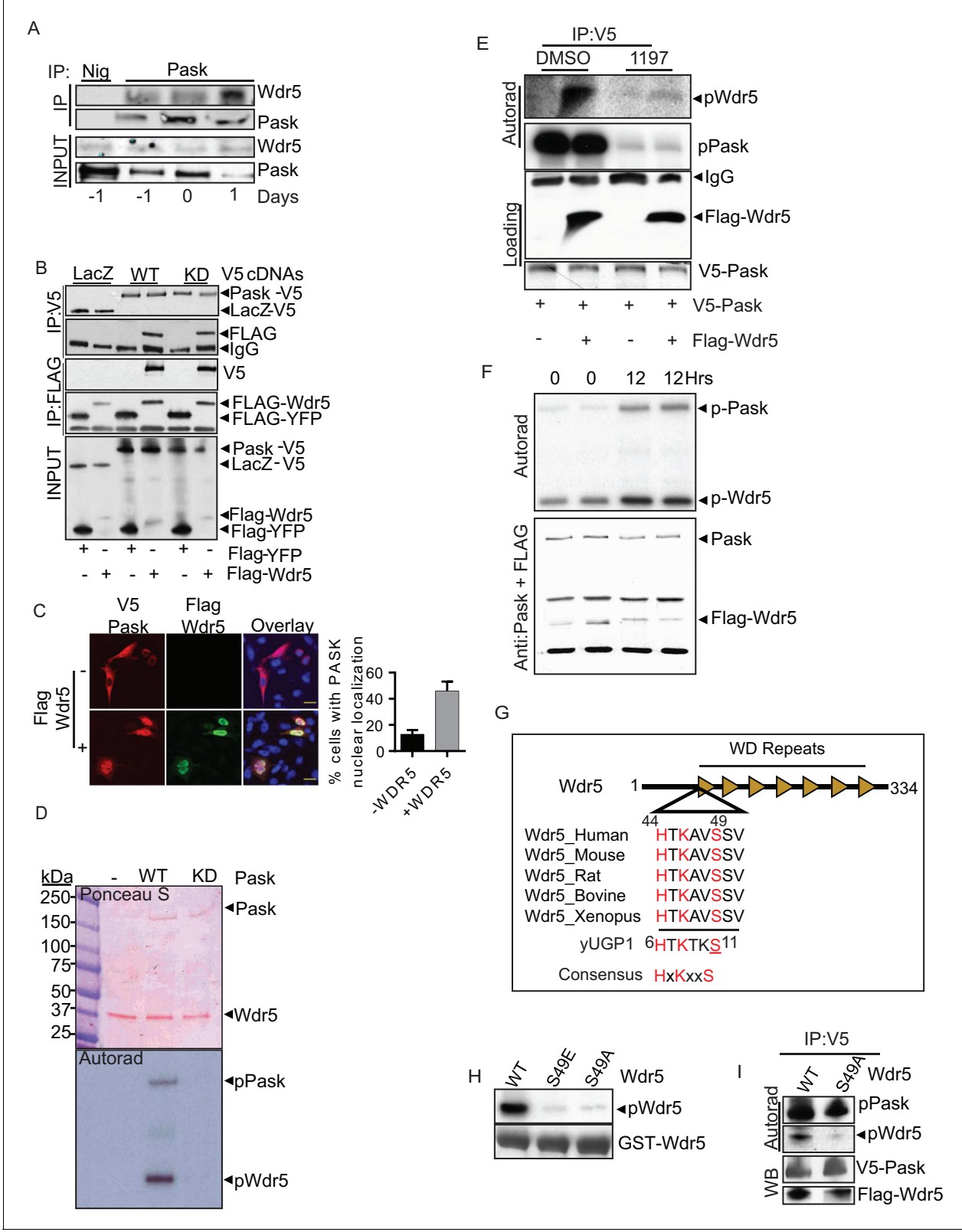

**Figure 4.** Pask directly interacts with and phosphorylates Wdr5 at Ser49. (**A**) Endogenous Pask was immunoprecipitated from C2C12 cells, either before (day -1, 0) or after induction of differentiation (Day 1). Immunoprecipitates were analyzed by western blot for Pask and Wdr5, indicating an enrichment of co-immunoprecipitation at Day 1 of differentiation. (**B**) V5-tagged LacZ, WT Pask or KD (K1028R) Pask was co-expressed with Flag-YFP or Flag-Wdr5 in 293T cells. V5 or Flag-tagged proteins were immunoprecipitated and examined by western blot using anti-Flag or V5 antibody. (**C**) V5-hPask was

*Figure 4 continued on next page*

eLIFE Research article

Biochemistry | Developmental Biology and Stem Cells

*Figure 4 continued*

expressed in HEK293T cells with control or Flag-Wdr5 vector. V5 and Flag were stained using Alexa Flour 568 or Alexa Flour 488, respectively. The fraction of cells with nuclear Pask localization was scored as a function of the presence (+) or absence (−) of Wdr5. (**D**) In vitro phosphorylation of purified His-Wdr5 was performed using WT or KD Pask and analyzed by autoradiogram of the reaction mixture after western blotting, with total protein visualized by Ponceau S staining. pPask indicates autophosphorylation of WT-Pask during kinase reaction. (**E**) The Pask-Wdr5 complex was immunoprecipitated from cells incubated with $^{32}$P in the presence of DMSO or 25 μM BioE-1197. Immunoprecipitates were analyzed by SDS-PAGE and autoradiography or western blot. (**F**) Endogenous Pask was immunoprecipitated from C2C12 cells growing in growth media or 12 hr after replacement with differentiation media containing 10 nM Insulin and was incubated with purified Flag-Wdr5 and [γ-$^{32}$P] ATP. Autoradiogram shows incorporation of $^{32}$P into Pask (p-Pask) and Wdr5 (p-Wdr5). (**G**) Schematic showing Ser49 and upstream sequence in Wdr5, compared to the site of Pask phosphorylation in Ugp1, a *bona fide* substrate of *S. cerevisiae* Pask. (**H**) GST-tagged WT, S49A or S49E Wdr5 was incubated with Pask and [γ-$^{32}$P] ATP and phosphorylation was detected by autoradiography after SDS-PAGE. (**I**) WT or S49A Wdr5 was co-immunoprecipitated with Pask from cells incubated with $^{32}$P-phosphate and analyzed as in (**E**).

The following figure supplement is available for figure 4:

**Figure supplement 1.** Pask directly interacts with and phosphorylates Wdr5 at Ser49.

*supplement 1E*). A very similar pattern was observed in co-immunoprecipitation of endogenous Pask with Wdr5[WT], WDR[S49A] and Wdr5[S49E] from cells (*Figure 4—figure supplement 1F*). We found that the association with endogenous Pask is weakened with Wdr5[S49E] and strengthened with Wdr5[S49A] compared with Wdr5[WT]. These data are consistent with the common observation that an unphosphorylated substrate (mimicked by the Ala mutant) engages with its kinase with higher affinity compared to its phosphorylated forms (mimicked by the Glu mutant). When examining components of the Mll1-4, SET1 and KAT2a complexes, we found that Wdr5[WT], WDR[S49A] and Wdr5[S49E] all bound to each member of these complexes equivalently, suggesting that Wdr5 phosphorylation by Pask does not alter recruitment of Wdr5 into its many chromatin-modifying complexes.

## Wdr5[S49E] restores *Myog* expression and myogenesis in the absence of Pask activity

Having established Wdr5 as a Pask substrate and given the published role of Wdr5-containing complexes in myoblast differentiation via the regulation of myogenic gene expression (*Asp et al., 2011*; *McKinnell et al., 2008*; *Segales et al., 2014*), we wanted to determine whether Wdr5 phosphorylation is the mechanism through which Pask regulates *Myog* expression and myocyte differentiation. If so, one might expect that expression of a Wdr5 variant that constitutively mimicked the phosphorylated form would reverse the effects of Pask inactivation and rescue the differentiation defect. Indeed, we found that expression of Wdr5[S49E], but not Wdr5[WT] or Wdr5[S49A], rescued the morphological features of C2C12 differentiation, including multinucleated myofibers (*Figure 5—figure supplement 1*). We next examined the ability of Wdr5[WT], Wdr5[S49A] and Wdr5[S49E] to rescue the expression of *MyoG* and its targets upon Pask knockdown. Wdr5[WT] and Wdr5[S49A] had minimal effect on *Pax7*, *Myog*, *Mylpf*, or *Acta1* mRNA abundance in Pask-siRNA cells (*Figure 5A*). On the other hand, the Wdr5[S49E] mutant either partially or completely rescued the defects in the expression of each of these genes caused by *Pask* knockdown. Wdr5[S49E] caused a modest increase in *Pask* expression, likely due to the effects of increased MyoG activity on the *Pask* promoter (*Figure 2—figure supplement 2F–G*).

Using microscopy of co-immunostained cells, we next examined the ability of GFP, Wdr5[WT], Wdr5[S49A] or Wdr5[S49E] to rescue the defect in progression from the Pax7[+] to MyoG[+] state in Pask-siRNA cells. GFP, Wdr5[WT], and Wdr5[S49A] all had no significant effect on the number of Pax7[+] cells (*Figure 5B*; quantified in *Figure 5C*, *Figure 5—source data 1*). Wdr5[S49E], on the other hand, completely reversed the aberrant increase in Pax7[+] cells caused by *Pask* knockdown. Importantly, this decrease in Pax7 positivity was only present in those cells in the population that expressed Wdr5[S49E] (S49E[+]; *Figure 5B*, indicated by arrows), and not in those cells in the population that were uninfected (S49E[-]). This demonstrates that these are cell autonomous effects that are directly and specifically related to the expression of the phospho-mimetic Wdr5 mutant. Those cells expressing Wdr5[S49E] also exhibited an almost complete rescue of MyoGexpression, while those expressing GFP and Wdr5[S49A] were unaffected (*Figure 5D*; quantified in *Figure 5E*). In this case, Wdr5[WT] also

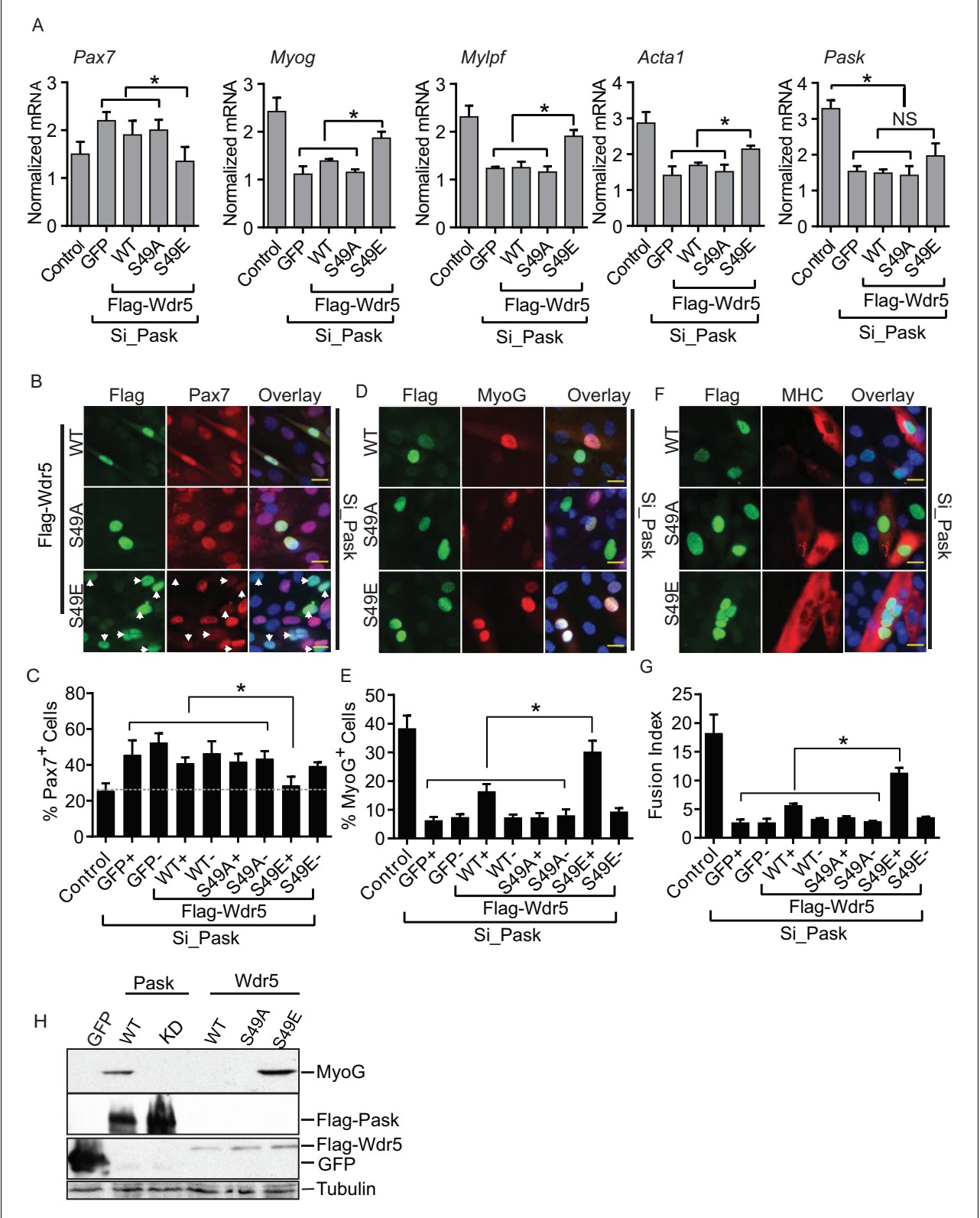

**Figure 5.** The phospho-mimetic S49E Wdr5 mutant rescues myogenesis in Pask-silenced cells. (**A**) GFP or WT, S49A or S49E Wdr5 were retrovirally expressed in Pask-siRNA C2C12 cells. qRT-PCR analysis was performed for the indicated mRNA on day 3 of differentiation. 18S rRNA was used as normalizer. n = 3. Error bars ± S.D *p<0.05. (**B**) Flag-tagged WT, S49A or S49E Wdr5 was expressed in Pask-siRNA C2C12 cells. After 24 hr, cells were stained for Pax7 at Day 0 of differentiation. Arrows show Wdr5$^{S49E}$-expressing cells and the corresponding cell autonomous decrease in Pax7

*Figure 5 continued on next page*

*Figure 5 continued*

expression. Scale bar = 20 μM. (**C**) Quantification of percent Pax7[+] cells from (**B**) as a function of the presence (+) or absence (−) of GFP or Wdr5. n = 3 independent experiments each with 100 cells counted. Error bars ± S.D. *p<0.05. (**D**) As in (**B**), except cells were stained for MyoG on Day 1 of differentiation. (**E**) Quantification of percent MyoG[+] cells from (**D**) as a function of the presence (+) or absence (−) of GFP or Wdr5. n = 3 independent experiments each with 100 cells counted. Error bars ± S.D. *p<0.05. (**F**) As in (**B**), except cells were stained for MHC on Day 3 of differentiation. (**G**) Quantification of fusion index from (**F**) as a function of the presence (+) or absence (−) of GFP or Wdr5. n = 3 independent experiments each with 100 cells counted. Error bars ± S.D. *p<0.05. (**H**) C2C12 myoblasts were infected with retrovirus expressing GFP, Flag tagged WT or KD Pask or WT, S49A or S49E Wdr5 and infected cells were selected with puromycin in growth media for 48 hr. Cells were lysed after selection and abundance of the indicated proteins was determined by Western blotting.

The following source data and figure supplement are available for figure 5:

**Source data 1.** Quantification of the Pax7+ and MyoG+ cell numbers and the fusion index in Wdr5S49E expressing cells.

**Figure supplement 1.** Wdr5[S49E] expression rescues genetic loss of Pask.

caused a significant increase in the fraction of MyoG[+] cells. As before, MyoG expression was unaffected in those cells in the Wdr5[WT] and Wdr5[S49E]-expressing populations that were uninfected. The defect in MHC expression and myotube formation caused by *Pask* knockdown was also robustly rescued in Wdr5[S49E]-expressing cells and weakly rescued in Wdr5[WT]-expressing cells (*Figure 5F,G*, *Figure 5—source data 1*).

Furthermore, we noticed that ectopic expression of Wdr5[S49E], like WT Pask, was sufficient to induce expression of MyoG even in the absence of differentiation stimuli (*Figure 5H*). Expression of kinase-dead (KD) Pask, Wdr5[WT] or Wdr5[S49A] failed to induce MyoG expression (*Figure 5H*) in the same growth media condition. Taken together, these data suggest that Wdr5 phosphorylation is a major mechanism whereby Pask promotes MyoG expression and myotube formation.

## Pask phosphorylation of Wdr5 promotes transcriptional derepression of the *Myog* promoter via H3K4me1 to H3K4me3 conversion and MyoD recruitment

At the onset of differentiation, the *Myog* promoter is remodeled with H3K4me3 modification resulting in its transcriptional activation (*Asp et al., 2011*; *Cheng et al., 2014*). Because transcriptional induction of *Myog* is dependent on Pask kinase activity or phospho-mimetic Wdr5[S49E], we tested whether Pask might act on the *Myog* promoter through regulating H3K4me3 accumulation. The *Myog* locus contains a region of H3K4me3 approximately -150 bp upstream of the transcriptional start site (TSS), which overlaps with the peak of MyoD binding marked by E-Box sequences (*Figure 6A*). In control cells, H3K4me3 abundance at two sites within this region of the promoter (*Myog_b* and *Myog_c*) was strongly increased on Day 1 of differentiation, while a region more distal to the TSS showed little H3K4me3 accumulation (*Myog_a*, *Figure 6B*). This increase in H3K4me3 was markedly blunted (*Myog_b*) or abolished (*Myog_c*) by *Pask* knockdown or inhibition (*Figure 6B*). In contrast, *Pask* knockdown had no effect on H3K4me3 abundance on the *Myod* promoter (*Figure 6B*, *Figure 6—source data 1*).

To assess whether these *Pask* knockdown phenotypes were *bona fide* and related to Wdr5 phosphorylation at Ser49, we tested the ability of WT or kinase dead (KD) hPask as well as WT, S49A or S49E variants of Wdr5 to rescue these effects. As we observed previously in microscopy experiments, expression of WT, but not KD, hPask completely rescued H3K4me3 occupancy of the *Myog* promoter (*Figure 6C*). Moreover, expression of Wdr5[S49E] restored H3K4me3 occupancy, whereas Wdr5[S49A] and Wdr5[WT] had no or modest effects, respectively (*Figure 6C*). One potential mechanism whereby Wdr5 phosphorylation might stimulate H3K4me3 occupancy of the *Myog* promoter is through enhanced Wdr5 promoter recruitment. To test this hypothesis, we performed Flag-Wdr5 ChIP-qPCR analysis on these same samples to determine Flag-Wdr5 occupancy on the *Myog* promoter. Consistent with that hypothesis, Wdr5[WT] modestly occupied the *Myog* promoter and this was decreased for the S49A mutant and increased for the S49E mutant (*Figure 6D*). Neither WT nor KD Pask occupied the *Myog* promoter (*Figure 6D*), consistent with our observation that Pask is not a stable member of any Wdr5-containing complexes. We also examined the Pask-dependence of

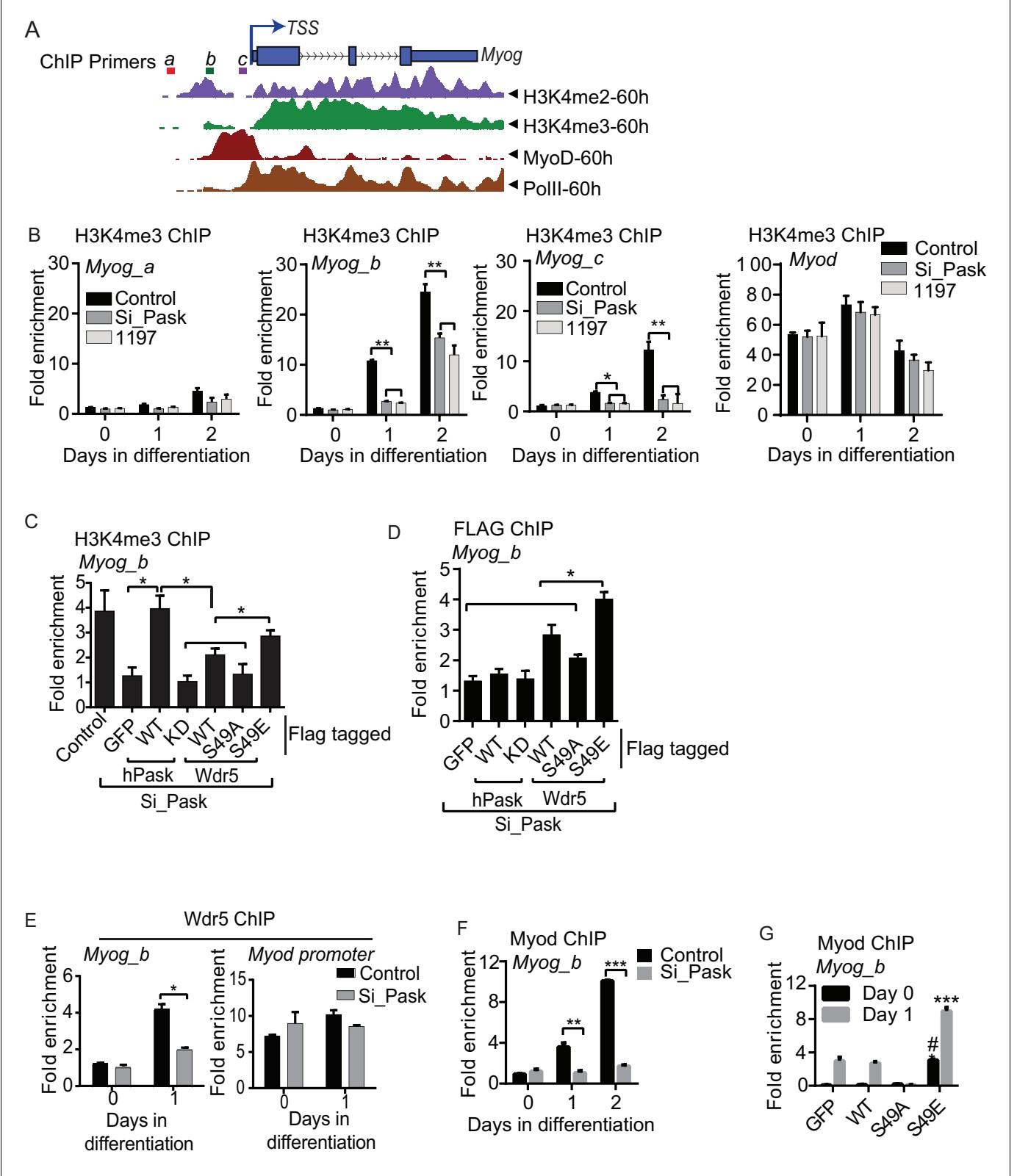

**Figure 6.** Pask is required for recruitment of Wdr5 and MyoD to the *Myog* promoter during differentiation. (**A**) A depiction of the *Myog* genomic locus depicting MyoD and RNAPolII occupancy as well as H3K4me2 and H3K4me3 abundance at 60 hr of differentiation from the ENCODE dataset for the C2C12 cell line. *TSS:* Transcription Start Site. Colored horizontal bars indicate the positions of ChIP amplicons *a, b* or *c*. (**B**) Fold H3K4me3 enrichment in control, Pask-siRNA or BioE-1197-treated C2C12 cells was assessed by ChIP-qPCR of the indicated amplicon followed by normalization against

*Figure 6 continued on next page*

*Figure 6 continued*

H3K4me3-deficient negative control region of the *actb* gene. n = 3. Error bars ± S.D. *p<0.05, **p<0.005. Because the *b* amplicon showed the most significant enrichment of H3K4me3 in control samples, it was selected for future studies. (C) H3K4me3 ChIP was performed from control or Pask silenced C2C12 cells expressing GFP, WT or KD Pask or WT, S49A or S49E Wdr5 at Day 1 of differentiation. n = 3. Error bars ± S.D. *p<0.05. (D) Flag ChIP was performed from Pask silenced C2C12 cells expressing GFP, WT or KD Pask or WT, S49A or S49E Wdr5 at Day 1 of differentiation. n = 3. Error bars ± S.D. *p<0.05. (E) Endogenous Wdr5 ChIP was performed from control or Pask-siRNA C2C12 cells at Day 0 or Day 1 of differentiation and fold enrichment on the *Myog* promoter was determined by qRT-PCR. Error bars ± S.D. *p<0.05. (F) MyoD ChIP was performed from control or Pask-siRNA C2C12 cells at Day 0, Day 1 or Day 2 of differentiation and fold enrichment of MyoD occupancy on the *Myog* promoter was determined by qRT-PCR. n = 3. Error bars ± S.D. **p<0.05, ***p<0.0005. (G) MyoD ChIP was performed from proliferating (Day 0) or differentiating (Day 1) C2C12 cells expressing GFP, Wdr5^WT^, Wdr5^S49A^ or Wdr5^S49E^. n = 3. Error bars ± S.D. #, ***p<0.0005, Wdr5 ^S49E^ vs GFP, Wdr5^WT^ or Wdr5^S49A^ at Day 0 and Day 1 respectively.

The following source data is available for figure 6:

**Source data 1.** Numerical values from the ChIP analysis represented in *Figure 6*.

endogenous Wdr5 occupancy of the *Myog* promoter and found that the increased binding of Wdr5 to the *Myog* promoter in response to differentiation cues observed in control cells was blunted in Pask-siRNA cells (*Figure 6E*). Wdr5 binding to the *Myod* promoter, on the other hand, was neither significantly induced upon differentiation, nor was it affected by *Pask* knockdown (*Figure 6E*), consistent with the observation that Pask is not required for expression of MyoD (*Figure 2B*, *Figure 2—figure supplement 2D–E*).

These data suggest that Wdr5 phosphorylation affects H3K4 trimethylation activity at the *Myog* promoter. MyoD is recruited to the *Myog* promoter during differentiation and then in turn orchestrates the chromatin remodeling process to promote transcriptional activation of *Myog* (*Palacios and Puri, 2006*; *Rampalli et al., 2007*; *Saccone and Puri, 2010*). We examined whether Pask plays a role in this process by assessing whether MyoD occupancy on the *Myog* promoter during differentiation was affected by *Pask* knockdown. Indeed, we found that *Pask* knockdown completely blocked the differentiation-stimulated increase in MyoD occupancy of the *Myog* promoter (*Figure 6F*, *Figure 6—source data 1*). This requirement for Pask is due to Wdr5 phosphorylation as expression of Wdr5^S49E^ increased MyoD occupancy of the *Myog* promoter at Day 1 of differentiation, whereas Wdr5^S49A^ blocked it (*Figure 6G*, *Figure 6—source data 1*). Interestingly, Wdr5^S49E^ increased the MyoD occupancy of the *Myog* promoter even in the absence of differentiation cues (Day 0) to a level similar to that found in control cells at Day 1 of differentiation (*Figure 6G*, *Figure 6—source data 1*). This result is consistent with our observation that Wdr5^S49E^ expression is sufficient to induce MyoG expression in the absence of differentiation cues (*Figure 5H*). Thus, it appears that Pask and/or phospho-Wdr5 are necessary and sufficient to stimulate the myogenic transcriptional cascade, and perhaps that MyoD occupancy and H3K4me3 modification synergize to maximize expression from the *Myog* promoter during differentiation.

The *Myog* promoter was recently shown to have high H3K4me1 occupancy in non-differentiating C2C12 cells (*Cheng et al., 2014*). These marks are converted to H3K4me3 marks in response to differentiation cues through an unknown mechanism (*Cheng et al., 2014*). Pask is activated by these same differentiation cues and expression of either WT Pask or phospho-mimetic Wdr5 is sufficient to bypass the requirement of differentiation cues. Hence, we asked whether Pask-Wdr5 signaling regulates H3K4me1 to H3K4me3 conversion on the *Myog* promoter in response to differentiation cues. In control cells, H3K4me1 marks were progressively depleted from the *Myog* promoter over the differentiation time-course (*Figure 7A*, *Figure 7—source data 1*), most likely due to further methylation to H3K4me3 (see *Figure 6B*). *Pask* knockdown severely blunted that loss of H3K4me1, in concert with blunting the appearance of H3K4me3 (*Figure 7A*, *6B*). This H3K4me1 to H3K4me3 conversion was accompanied by a decrease in total histone H3 density at the *Myog* promoter in control cells (*Figure 7B*), suggestive of a transition from a transcriptionally inactive or closed (nucleosome rich) to a transcriptionally accessible or open (nucleosome depleted) state that might facilitate MyoD recruitment in control cells. This loss of H3 density during differentiation was abolished by *Pask* knockdown (*Figure 7B*, *Figure 7—source data 1*). Because H3K4me1 is commonly found at enhancers, we also asked whether Pask is required for H3K4me1 occupancy at the *Myog* enhancer.

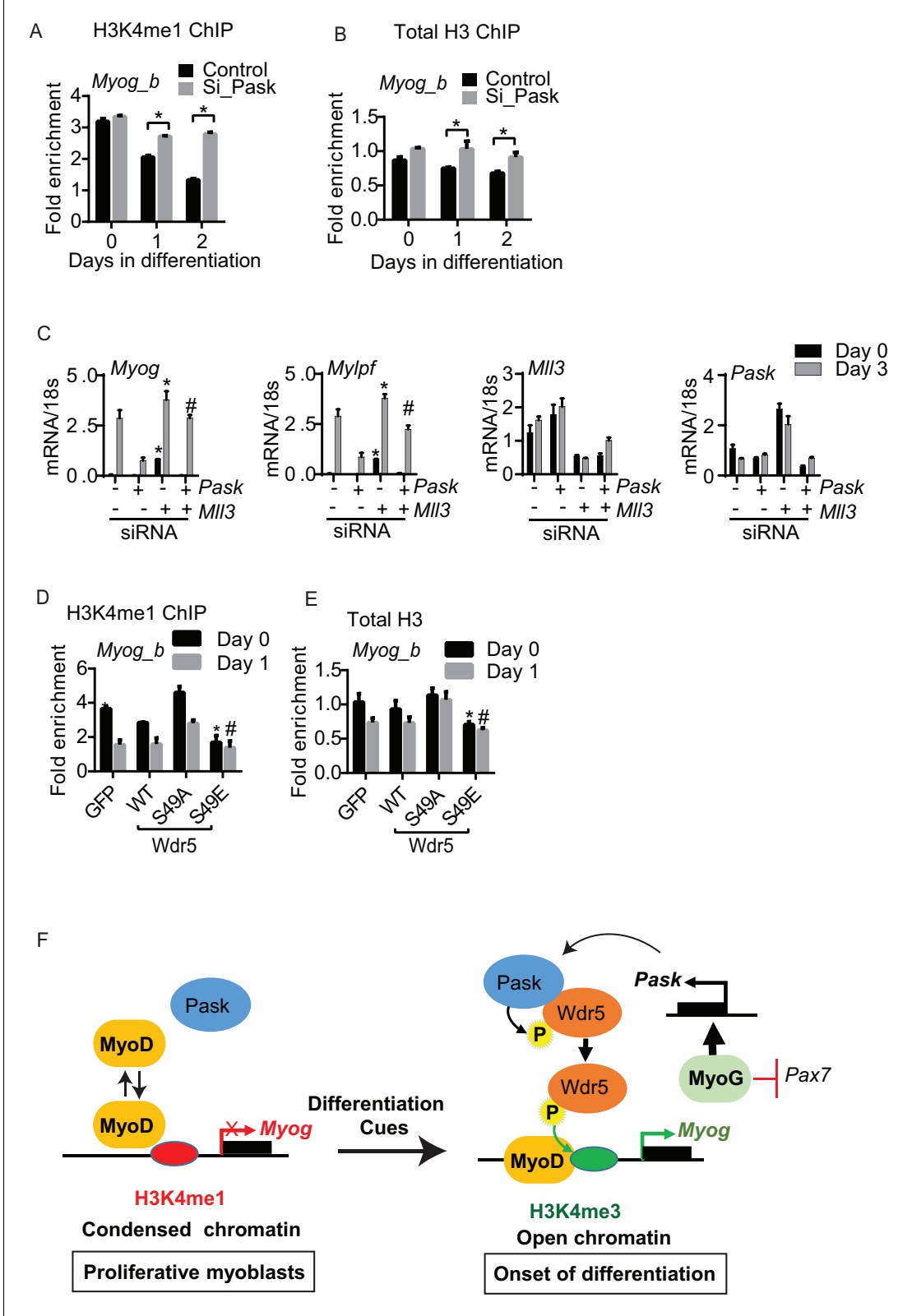

**Figure 7.** Differentiation induced H3K4me1 to H3K4me3 conversion is dependent upon Pask phosphorylation of Wdr5. (A) H3K4me1 and (B) total H3 ChIP were performed from control or Pask-siRNA C2C12 cells at the indicated day of differentiation and fold enrichment on the *Myog* promoter was determined by qRT-PCR using primer set *b*. Error bars ± S.D. *p<0.05, **p<0.005. (C) The differentiation potential of C2C12 myoblasts subjected to *Pask* or *Mll3* siRNA treatments was assessed by qRT-PCR using primers specific for *Myog, Mylpf, Mll3 and Pask*. (D) H3K4me1 or (E) total H3 ChIP was

eLIFE Research article

Biochemistry | Developmental Biology and Stem Cells

*Figure 7 continued*

performed from proliferating (Day 0) or differentiating (Day 1) C2C12 cells expressing GFP, Wdr5$^{WT}$, Wdr5$^{S49A}$ or Wdr5$^{S49E}$. n = 3. Error bars ± S.D. *, #p<0.05, Wdr5 $^{S49E}$ vs GFP, Wdr5$^{WT}$ or Wdr5$^{S49A}$ at Day 0 and Day 1 respectively. (**F**) Model depicting the role of Pask and Wdr5 phosphorylation in regulating MyoD recruitment to the *Myog* promoter during differentiation. See Discussion for detail.

The following source data and figure supplement are available for figure 7:

**Source data 1.** Numerical values from the ChIP analysis represented in *Figure 7*.

**Figure supplement 1.** Pask and phosphomimetic Wdr5 promote H3K4me1 to H3K4me3 conversion and MyoD recruitment to the Myog promoter.

However, unlike the proximal *Myog* promoter, H3K4me1 to H3K4me3 conversion at the *Myog* upstream enhancer is not impaired by *Pask* knockdown, nor is the appearance of H3K27ac (***Figure 7—figure supplement 1B***). Interestingly, MyoD recruitment to the enhancer during differentiation is impaired similarly to the proximal promoter by *Pask* knockdown, possibly due to the interdependence of the enhancer and promoter for optimal activation (***Sanyal et al., 2012***). Like *Myog,* the *Mybph* and *Acta1* promoters, but not *Myod* were reported to undergo H3K4me1 to H3K4me3 conversion during differentiation in C2C12 myoblasts (***Cheng et al., 2014***) and this is also Pask-dependent (***Figure 7—figure supplement 1C,D***).

If the enhanced H3K4me1 occupancy at the *Myog* promoter is responsible for the transcriptional repression observed upon loss of Pask activity, then silencing of *Mll3* at the onset of differentiation, which is specific for deposition of H3K4me1 marks might rescue the expression of *Myog* upon *Pask* knockdown. Indeed, we found that *Mll3* knockdown in Pask-siRNA cells rescued the impaired *Myog* expression and restored differentiation as indicated by the *Mylpf* expression (***Figure 7C***). Taken together, our results show that Pask is an essential mediator whereby differentiation cues stimulate H3K4me1 to H3K4me3 conversion and chromatin remodeling to facilitate MyoD recruitment to the *Myog* promoter.

The requirement of Pask for loss of H3K4me1 and total H3 from the *Myog* promoter appears to be related to Wdr5 phosphorylation since Wdr5$^{S49E}$ expression was sufficient to deplete H3K4me1 marks (***Figure 7D***) and total H3 occupancy (***Figure 7E***) on the *Myog* promoter even in the absence of differentiation cues. In contrast, the depletion of H3K4me1 and total H3 at Day 1 of differentiation was prevented by the expression of Wdr5$^{S49A}$ (***Figure 7D,E***). These effects of Wdr5$^{S49A}$ and Wdr5$^{S49E}$ expression mirror those on MyoD occupancy of the *Myog* promoter (***Figure 6G***). Thus, Wdr5 phosphorylation status appears to dictate MyoD recruitment to the *Myog* promoter, perhaps via H3K4me1 to H3K4me3 conversion.

## Discussion

We demonstrate herein, using multiple genetic and pharmacologic manipulations, that Pask is required for the differentiation of ES cells, adipogenic stem cells and myoblasts. Focusing on myoblast differentiation as a powerful and facile model system, we show that Pask phosphorylation of Wdr5 promotes several molecular events that together enable the expression of *Myog*, which encodes the critical myogenic transcription factor Myogenin. Pask activity or expression of Wdr5$^{S49E}$, which mimics the Pask phosphorylated form of Wdr5, is necessary and sufficient to drive H3K4me1 to H3K4me3 conversion, MyoD recruitment and transcription at the *Myog* promoter. Based on the data presented herein, we propose a model for C2C12 cells wherein under proliferative conditions, MyoD is unable to bind stably to the *Myog* promoter due to a repressive chromatin configuration enforced by H3K4me1 marks (***Figure 7F***). As a result, *Myog* is not expressed and differentiation is prevented. Differentiation cues, including insulin, stimulate Pask kinase activity, Pask-Wdr5 association and Pask phosphorylation of Wdr5 at Ser49. Phosphorylation of Wdr5, which is a component of histone methylation complexes, facilitates the conversion of H3K4me1 to H3K4me3 at the *Myog* promoter resulting in chromatin remodeling, stable MyoD recruitment, transcription initiation and establishment of the differentiation program. Once this program initiates, it is buttressed by MyoG, likely in collaboration with MyoD, stimulating *Pask* expression while suppressing *Pax7* expression

(*Olguin et al., 2007*). This self-reinforcing transcriptional circuit, driven by interlocking positive and negative feedback loops, creates stable commitment to the myogenic cell fate (*Figure 7F*).

In spite of the role of *Pask* in the muscle regeneration response to injury in vivo that we have described, *Pask*^-/-^ mice are developmentally normal. While surprising, this is similar to the phenotype of mice lacking the key myogenic transcription factors, Myod (*Rudnicki et al., 1992*), Pax7 (*Seale et al., 2000*), Myf5 (*Kassar-Duchossoy et al., 2004*; *Kaul et al., 2000*) or Mrf4 (*Kassar-Duchossoy et al., 2004*) (reviewed in (*Musarò and 2014*). Moreover, we found that myoblasts from *Pask*^-/-^ mice had acquired resistance to Pask inhibitor-mediated suppression of myocyte differentiation, indicating that constitutive knockout results in a pro-differentiation compensatory response that is Pask-independent. Similarly, *Pask*-siRNA or PASK-inhibited cells typically exhibited modest induction of *Myog*, its target genes and H3K4me3 abundance upon differentiation, suggesting the presence of *Pask*-independent mechanisms to drive myogenin induction. It's noteworthy that *Pask*^-/-^ myoblasts exhibit elevated expression of *Myf5* (*Figure 2B*), which has been shown to be able to compensate for the loss of MyoD function (*Conerly et al., 2016*; *Rudnicki et al., 1992*). Thus, it is possible that Myf5 and other transcriptional regulators drive some of the Pask-independent activation of the *Myog* promoter. Nonetheless, our data demonstrate that the Pask-Wdr5 signaling axis is necessary and sufficient for full activation of the *Myog* promoter and subsequent myoblast fusion.

## Transcriptional regulation of Pask in stem and progenitor cells

We have shown that *Pask* expression is high in stem and progenitor cell types (*Figure 1—figure supplement 1A–C*). In ES cells, the *Pask* promoter is occupied by the pluripotency promoting transcription factors, Oct4 and Nanog (*Figure 1—figure supplement 1D*). This transcription factor occupancy coincided with the *Pask* transcriptional activation as indicated by the occupancy of RNA PolII as well as the activating H3K4me3 and H3K27ac chromatin modifications. During muscle regeneration in vivo expression is dramatically induced as early as 3 days after injury (*Figure 1H*), coincident with expansion of the satellite cell population and stimulation of MyoG expression (*Braun and Gautel, 2011*). We showed that MyoG and MyoD bind the *Pask* promoter (*Figure 2—figure supplement 3A–B*), which is marked by broad domains of H3K4me3 occupancy similar to those found in other lineage-specifying genes (*Benayoun et al., 2014*). Terminal differentiation of ES cells and C2C12 myoblasts results in a decline in *Pask* expression, which ultimately reaches a low steady-state abundance when differentiation is complete. This pattern is consistent with the expression of other key regulators of terminal differentiation like MyoD and hints at the importance of Pask in the lineage commitment process.

## Pask and Wdr5 phosphorylation promote terminal differentiation

Pharmacologic inhibition of Pask activity caused a loss of terminal differentiation in three differentiation paradigms: ES cells to a neuronal fate, C3H10T1/2 mesenchymal stem cells to adipocytes and C2C12 myoblasts to myotubes. Utilizing the myoblast differentiation system, we have identified and described Wdr5 as a Pask substrate that mediates these differentiation effects. Wdr5 has been previously implicated in controlling stem cell maintenance. For example, Oct4 associates with Wdr5 and recruits H3K4me3 complexes to its target promoters to induce transcriptional activation and maintenance of pluripotency (*Ang et al., 2011*). In C2C12 myoblasts, Wdr5 was shown to interact with Pax7 to enforce myogenic specification (*Kawabe et al., 2012*; *McKinnell et al., 2008*; *Rudnicki et al., 2008*). Our results provide a new regulatory role for Wdr5 in promoting differentiation via the expression of key target genes. More importantly, these results define a novel regulatory paradigm wherein the function of Wdr5 is modulated by Pask-dependent phosphorylation. While we deem it likely, it remains to be established whether Wdr5 phosphorylation is a conserved and required mechanism whereby Pask promotes the differentiation of other stem and progenitor cells.

## Pask as a signaling intermediate in cell differentiation

Differentiation cues engage and activate the signaling pathways and transcriptional networks that combine to drive terminal differentiation (*Basson, 2012*). While a pioneering transcription factor like MyoD can occupy its target promoters in a sequence-specific manner, the full activation of the MyoD transcriptional response depends upon signaling from differentiation cues (*Berkes and Tapscott, 2005*). These cues, including those used in vitro treatment with insulin, are thought to enhance

MyoD binding to its target promoters and recruit histone modifying proteins to initiate the expression of differentiation genes (*Berkes and Tapscott, 2005*; *Blum and Dynlacht, 2013*; *Braun and Gautel, 2011*; *Dilworth and Blais, 2011*). Ectopic expression of Pask or phospho-mimetic Wdr5 was sufficient to induce all of these effects in the absence of differentiation cues (*Figure 2K–J*, *Figure 5H*). These data raise the possibility that Pask might be a critical node in the signaling network connecting differentiation cues with myogenic gene expression. Consistent with that hypothesis, we found that Pask activity was stimulated by differentiation cues as was the Pask-Wdr5 interaction (*Figure 4F*). We therefore believe that differentiation cues act, at least in part, to drive the myogenic transcriptional program via Pask activation and phosphorylation of Wdr5. It remains unknown how Pask activity and the Pask-Wdr5 interaction are stimulated by differentiation cues, but we suggest that these phenomena are likely central components of this signaling pathway.

### Role of Pask-pWdr5 in H3K4me1 to H3K4me3 conversion

H3K4me1 was recently demonstrated to be a transcriptionally repressive mark at the promoters of myogenic genes (*Cheng et al., 2014*). In response to differentiation cues, H3K4me1 is further methylated to H3K4me3 on these promoters to initiate their expression. H3K4 monomethylation is catalyzed by the Wdr5-containing Mll3/Mll4 complexes (*Hu et al., 2013*), while the Wdr5-containing Mll1/Mll2 complexes catalyze H3K4 trimethylation (*Shilatifard, 2012*). Thus, it has been predicted that differentiation cues promote a Mll3/Mll4 to Mll1/Mll2 switch on myogenic promoters via an unknown mechanism (*Cheng et al., 2014*). We speculate that Pask phosphorylation of Wdr5 might be required for this process. For example, Pask silencing prevented the H3K4me1 to H3K4me3 conversion on the *Myog* promoter in response to differentiation cues and silencing of Mll3 rescued the defect in the induction of *Myog* expression caused by *Pask* silencing. Interestingly, H3K4me1 and H3K4me3 was unaffected at the *Myod* promoter or at the *Myod* and *Myog* enhancers during differentiation in *Pask*-silenced condition. This suggests that Pask and Wdr5 phosphorylation are not required for the enzymatic activity of the Mll1/2 or Mll3/4 complexes, but only for their activity on specific promoters like *Myog, Acta1* and *Mybph*. These promoters are repressed during proliferation and required to be derepressed at the onset of differentiation. Our data suggest that Pask mediates chromatin remodeling including H3K4me1 to H3K4me3 conversion, resulting in enhanced MyoD binding and transcriptional activation during differentiation via Wdr5 phosphorylation. Future studies will determine whether Pask-mediated Wdr5 phosphorylation promotes switching of the Mll3/4 and Mll1/2 complexes at the *Myog* promoter.

Taken together, we have identified Pask and Wdr5 phosphorylation as necessary for differentiation of myocytes in vitro. Because the myocyte differentiation epigenetic program is similar to those found in other differentiation paradigms, including those we show to be Pask dependent, we speculate Pask likely functions through similar mechanisms in other systems. Specifically, we suggest that Pask phosphorylates Wdr5 to promote H3K4me1 to H3K4me3 conversion on the promoters of lineage-specifying genes to facilitate chromatin remodeling, gene expression and differentiation. As a known component of hormonal and nutrient signaling pathways, future studies of Pask functions in stem cells have the promise of opening exciting new avenues that combine metabolic signaling and stem cell fate determination with the control of terminal differentiation.

## Materials and methods

Please refer to resource table for resource ID of the reagents used in this study in supplementary information.

### Cell lines and differentiation paradigm

HEK 293T, C2C12 myoblasts and C3H10T1/2 mesenchymal stem cells were obtained from the American Type Culture Collection (ATCC). Human skeletal muscle primary myoblasts (Cat# SKB-F) were obtained from Zenbio labs Inc. These cell lines were verified for authenticity by ATCC and Zenbio labs and all cell lines were routinely tested to be free of mycoplasma using PlasmoTest (In vivogen Imc.). HEK293T, C2C12 and C3H10T1/2 cells were maintained in DMEM with 10% FBS (fetal bovine serum) and 1% PS (penicillin and streptomycin) (growth media, GM) at <60% confluency. Human skeletal muscle myoblasts were maintained in proprietary media obtained from Zenbio labs. Mouse embryonic cell culture conditions and differentiation were as described previously

(*Shakya et al., 2015*). C3H10T1/2 differentiation was as described previously (*Villanueva et al., 2011*) except that BioE-1197 or DMSO was added 48 hr prior to induction of the differentiation regime. C2C12 differentiation was initiated at 95% confluency by switching from GM to DMEM with 2% horse serum and 1%PS (DM) or DMEM + 10nM Insulin + 1% PS (Insulin differentiation media). Day 0 time-point samples were collected at 95% confluency before switching to DM. During differentiation, fresh DM was applied every 24 hr until the end of the experiment.

## iPSC generation

mSTEMCCA construct containing mouse Oct4, Sox2, Klf4 and c-Myc was obtained from R. Mostoslavsky. Lentivirus was produced by co-transfecting 293T cells with 1.7 µg each of packaging plasmids (pMDLg/pRRE,pRSV-Rev) and 1.7 µg envelope plasmid (pVSVG) and 5 µg mSTEMCCA. 1 × 10$^5$ Oct4-GFP MEFs generated from *Pou5f1*[tm2Jae/J] (Jackson Lab) were plated on the feeders in 6-well plates and were infected with lentivirus for 48 hr in the presence of 4 µg/ml polybrene (Sigma). The ESC medium with or without 50 µM BioE-1197 was changed every day. From day 8, GFP colonies were counted and images were captured with an Olympus IX51 inverted microscope.

## Plasmids and retroviral infection

Human WT and K1028R (KD) (*Kikani et al., 2010*) versions of Pask were cloned into the pQCXIP vector with N-terminal Flag tags. pCDNA FlagWdr5 (Addgene Cat#15552) and pGEX-5X1-Flag-Wdr5 (Addgene Cat#15553), pCDNA3 Flag Rbbp5 (Addgene Cat# 15550) and pCDNA3-Flag-PTIP (Addgene Cat # 15557) were deposited by Dr. Kai Ge (*Cho et al., 2007*). S49A and S49E mutations in pCDNA3 Flag-Wdr5 were made using sewing PCR-based mutagenesis in these vectors. For retroviral expression, WT, S49A and S49E Wdr5 were subcloned into pQCXIP vector. pQCXIP/GFP was described previously (*Chen et al., 2014*). pCL-Flag-PCAF (KAT2B, Addgene Cat#8941) was deposited by Dr. Yoshihiro Nakatani (*Yang et al., 1996*). pCDNA-Set9 (Addgene Cat#24084) was deposited by Dr. Danny Reinberg. pCDNA-Flag-Menin (Addgene Cat# 32079) was deposited by Dr. Matthew Meyerson. pCL-Babe-MyoD (Addgene Cat# 20917) was deposited by Dr. Stephen Tapscott (*Yang et al., 2009*). Rat MyoG was cloned into the pQCXIP vector after PCR from a rat cDNA library. Retroviral production for infection was as described (*Kikani et al., 2012*).

## Satellite cell isolation

Satellite cells were isolated from 10–12 weeks old WT and *Pask*[-/-] littermates according to published protocol (*Danoviz and Yablonka-Reuveni, 2012*). Briefly TA muscles from hind limbs of WT or Pask[-/-] mice were isolated, minced in DMEM and enzymatically digested with 0.1% Pronase for 1 hr. After repeated trituration, the cell suspension was filtered through 40 µM filter. Cells were plated on matrigel precoated plates and allowed to grow for four days. The differentiation of these satellite cells derived myoblasts was stimulated by the addition of 100 nM insulin in serum free DMEM.

## Pask siRNA knockdown and inhibition

siRNA-mediated gene knockdown was performed by transfecting 50nM of pooled siRNA against control (Cat #ID-001810-10-05), mouse Pask (L-065533-00-0005) or mouse Myod1 (L-041113-00-0005) purchased from Dharmacon using Lipofectamine RNAimax (Life Technologies Inc, Carlsbad, CA). Knockdown was carried out with the cells at 40% confluence in suspension. Cells were allowed to attach and grow for 48 hr when they reached 95% confluence, which marked the Day 0 point. Cells were either harvested at this point or differentiation was initiated by switching from GM to DM. For Pask inhibition by BioE-1197, C2C12 or human primary skeletal muscle myoblasts were seeded at 40% density in the presence of 25 µM BioE-1197 or equal volume of DMSO. Cells were allowed to grow for 48 hr at which point they reached 95% confluence and differentiation was initiated by switching from GM to DM in the presence of 25 µM BioE-1197 or DMSO.

## Quantitative reverse transcription PCR

Total RNA was prepared from C2C12 or human myoblasts subjected to differentiation conditions as described. qRT-PCR was performed from cDNA prepared from RNA using mRNA target-specific primers. Three independent experiments, done in triplicate with identical experimental parameters, were used for statistical analysis of mRNA transcript abundance. Student t-test was used for

statistical significance with significance value set to p<0.05. Sequences for qRT-PCR primers used in this study are available in resource table at the end of this section.

## Protein extracts, co-immunoprecipitation and western blot analysis

For quantitation of protein abundance during differentiation, cells were lysed in RIPA buffer, cellular debris was eliminated by centrifugation, and lysates were separated by SDS-PAGE. For co-immuno-precipitation, cells were lysed in native lysis buffer described previously (*Kikani et al., 2010*). Immu-noprecipitation was performed using the designated antibodies bound to Protein G beads (Pierce Cat# 22852). Protein complexes were washed with lysis buffer five times, denatured and separated by SDS-PAGE. Please refer to resource table for the list of antibodies used and their resource ID.

## Immunofluorescence microscopy

C2C12 cells growing on coverslips were fixed with 4% Paraformaldehyde and permeablized with 0.2% Triton-X 100. Following 1 hr of blocking with 10% normal goat serum, the indicated primary antibodies were added for overnight incubation at 4°C. Following three washes with ice cold PBS, cells were incubated with anti-mouse Alexa fluor 568 (for Pax7, MyoD, MyoG and MHC) or anti-rab-bit Alexa fluor 488 (Flag or Pask) secondary antibodies for 1 hr in dark at room temperature. The coverslips were mounted using Prolong-Anti-fade mounting media containing DAPI. Fusion index was used as a measure of differentiation and was calculated as the percent of nuclei in MHC$^+$ cells in relation with total nuclei. For quantification of microscopic images, at least 100 cells were counted from three separate experiments in a sample-blinded manner. Statistical significance was calculated using Student's t-test with p<0.05 set as the significance level.

## Muscle injury

Animal experiments were performed in accordance with protocols approved by the Institutional Ani-mal Care and Use Committee at the University of Utah to JR (16-05010) and GK (16-00074). Injury and subsequent analysis of skeletal muscle regeneration was performed using previously established methods (*Murphy et al., 2011*).

## In vitro and in situ phosphorylation of Wdr5

For in vitro phosphorylation of Wdr5, His- or GST-tagged Wdr5 proteins purified from *E. coli* were incubated with WT or KD Pask in the presence or absence of BioE-1197 in kinase reaction buffer containing [γ-$^{32}$P]ATP (*Kikani et al., 2010*). Kinase reactions were terminated after 10 min by the addition of SDS buffer. Proteins were separated by SDS-PAGE, transferred onto nitrocellulose and exposed to autoradiographic film. For in situ phosphorylation of Wdr5, metabolic $^{32}$P labeling was performed as described (*Kikani et al., 2010*) from cells co-expressing WT Pask with WT or S49A Wdr5. Incorporation of phosphate into Wdr5 was detected by purifying the Pask-Wdr5 com-plex from cells and determining phosphorylation by autoradiography.

## Chromatin Immunoprecipitation

Chromatin Immunoprecipitation (ChIP) from C2C12 cells was performed according to (*Hollenhorst et al., 2007*) with the following modifications. $1 \times 10^7$ C2C12 cells/15cm plate were treated for differentiation according to the procedures described above. At the appropriate time-point, cells were washed twice with PBS and cross-linked with 1% formaldehyde in PBS for 10 min at room temperature. The cross-linking was quenched by the addition of glycine to a 125 mM final con-centration. Nuclear pellets were sonicated using a Branson 450 Sonifier to prepare sheared chroma-tin extracts. Appropriate ChIP grade antibodies, as indicated in resource ID, were incubated with Dynabeads (Life Technologies Cat # 11202D – anti-Mouse or 11,204 – anti-Rabbit), to which sheared chromatin was added for immunoprecipitation. Crosslinks were reversed and DNA was purified using the Qiagen PCR purification kit. A sample with no immunoprecipitation step was processed in parallel as the input sample. qRT-PCR analysis of ChIP DNA was performed according to (*Hollenhorst et al., 2007*). Fold enrichment is calculated as the ratio of the signal of the target region over the signal of a negative control genomic region (~20,000 bp upstream of the *actb* gene). Sequences of primers for ChIP are available upon request.

## Statistical analysis

Data are presented as mean ± standard deviation (SD). Student's *t* test with -tailed equal variance with paired type analysis was used for calculating statistical significance between control and test sample. $p < 0.05$ is accepted as significant difference between control vs test sample. Experiments were performed in independent sets of triplicates.

## Additional information

### Funding

| Funder | Grant reference number | Author |
|---|---|---|
| National Institute of Diabetes and Digestive and Kidney Diseases | 5R01DK071962-08 | Jared Rutter |
| National Heart, Lung, and Blood Institute | Post Doctoral Training Fellowship | Chintan K Kikani |

The funders had no role in study design, data collection and interpretation, or the decision to submit the work for publication.

### Author contributions

CKK, Conception and design, Acquisition of data, Analysis and interpretation of data, Drafting or revising the article; XW, LP, HS, ZS, Acquisition of data, Analysis and interpretation of data; AS, Acquisition of data, Contributed unpublished essential data or reagents; AK, Acquisition of data, Analysis and interpretation of data, Drafting or revising the article; CV, BG, Drafting or revising the article, Contributed unpublished essential data or reagents; GK, Conception and design, Analysis and interpretation of data, Drafting or revising the article, Contributed unpublished essential data or reagents; DT, JR, Analysis and interpretation of data, Drafting or revising the article, Contributed unpublished essential data or reagents

### Author ORCIDs

Chintan K Kikani, http://orcid.org/0000-0003-1140-0192
Alexandra Keefe, http://orcid.org/0000-0001-9947-3403
Dean Tantin, http://orcid.org/0000-0003-1354-8385
Jared Rutter, http://orcid.org/0000-0002-2710-9765

### Ethics

Animal experimentation: Animal experiments were performed in accordance with protocols approved by the Institutional Animal Care and Use Committee at the University of Utah to JR (16-05010) and GK (16-00074).

## Additional files

### Major datasets

The following previously published datasets were used:

| Author(s) | Year | Dataset title | Dataset URL | Database, license, and accessibility information |
|---|---|---|---|---|
| Jane E Lattin, Kate Schroder, Andrew I Su, John R Walker, Jie Zhang, Tim Wiltshire, Kaoru Saijo, Christopher K Glass, David A Hume, Stuart Kellie, Matthew J Sweet | 2008 | Expression analysis of G Protein-Coupled Receptors in mouse macrophages | http://ds.biogps.org/?dataset=GSE10246&gene=269224 | Publicly Available at the NCBI Gene Expression Omnibus (Accession no: GSE10246) |

| | | | | |
|---|---|---|---|---|
| Loh YH1, Wu Q, Chew JL, Vega VB, Zhang W, Chen X, Bourque G, George J, Leong B, Liu J, Wong KY, Sung KW, Lee CW, Zhao XD, Chiu KP, Lipovich L, Kuznetsov VA, Robson P, Stanton LW, Wei CL, Ruan Y, Lim B, Ng HH | 2006 | The Oct4 and Nanog transcription network that regulates pluripotency in mouse embryonic stem cells | http://www.ncbi.nlm.nih.gov/geo/query/acc.cgi?acc=GSE4189 | Publicly Available at the NCBI Gene Expression Omnibus (Accession no: GSE4189) |
| Ohi Y, Qin H, Hong C, Blouin L, Polo JM, Guo T, Qi Z, Downey SL, Manos PD, Rossi DJ, Yu J, Hebrok M, Hochedlinger K, Costello JF, Song JS, Ramalho-Santos M | 2011 | Incomplete DNA methylation underlies a transcriptional memory of somatic cells in human iPS cells | http://www.ncbi.nlm.nih.gov/geo/query/acc.cgi?acc=GSE23034 | Publicly Available at the NCBI Gene Expression Omnibus (Accession no: GSE23034) |
| Cao F, Wagner RA, Wilson KD, Xie X, Fu J, Drukker M, Lee A, Li RA, Gambhir SS, Weissman IL, Robbins RC, Wu JC | 2008 | Transcriptional Profiling of Human Embryonic Stem Cell-Derived Cardiomyocytes | http://www.ncbi.nlm.nih.gov/geo/query/acc.cgi?acc=GSE13834 | Publicly Available at the NCBI Gene Omnibus (Accession no: GSE13834) |

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
