## [Decision Letter]

Thank you for submitting your article "PASK integrates hormonal signaling with histone modification via Wdr5 phosphorylation to drive myogenesis" for consideration by *eLife*. Your article has been favorably evaluated by K VijayRaghavan (Senior Editor) and three reviewers, one of whom, Peter Tontonoz, is a member of our Board of Reviewing Editors. The following individuals involved in review of your submission have agreed to reveal their identity: Bradley B Olwin (Reviewer #2); Elizaveta V Benevolenskaya (Reviewer #3).

The reviewers have discussed the reviews with one another and the Reviewing Editor has drafted this decision to help you prepare a revised submission.

Summary:

In the manuscript the authors provide a comprehensive analysis of the role for PASK in regulating satellite cell fate decisions. The authors find that PASK via phosphorylation of Wdr5 alters methylation of the Myog gene promoter altering accessibility for MyoD, initiating a feed-forward loop required for Myogenin induction irreversibly committing the myoblast to terminal differentiation. The data are thorough, comprehensive and the approach elegant and convincing. The data provide a significant step forward in understanding commitment to terminal differentiation and will have a major impact on research in skeletal muscle. If broadly applicable, the data will impact general stem cell research.

Essential revisions:

1) The authors claim that PASK is dispensable for iPSC reprograming but necessary for differentiation. Several prior studies have suggested that WDR5 is important for self-renewal, proliferation and reprogramming through promotion of H3K4me3, and that loss of WDR5 function leads to activation of genes that promote differentiation (Ang et al. Cell 2011, Yang et al. *eLife* 2014, Wang et al. Nature 2011). This issue should at least be addressed in the Discussion.

2) Are there any effects on proliferation rate due to loss of WDR5 phosphorlyation under proliferative culture conditions?

3) PASK knockdown had no effect on H3K4me3 signature at the MyoD promoter (or expression of MyoD) although it did influence H3K4me3 and MyoD occupancy at MyoG promoter to drive the expression of MyoG in response to differentiation cues. While MyoD is required for the driving myogenesis, is it specifically necessary for PASK dependent changes in trimethylation of H3K4 at the MyoG promoter? In other words, given the multiple positive reinforcing feedback loops in the proposed model, could it be that MyoD acts cooperatively with WDR5 to change chromatin accessibility? The hierarchical relationship between these factors can be clarified. For example, are there differences in the observed changes in chromatin signatures with pask inhibition or deletion upon MyoD knockdown?

---

## [Author Response]

*Essential revisions:*

*1) The authors claim that PASK is dispensable for iPSC reprograming but necessary for differentiation. Several prior studies have suggested that WDR5 is important for self-renewal, proliferation and reprogramming through promotion of H3K4me3, and that loss of WDR5 function leads to activation of genes that promote differentiation (Ang et al. Cell 2011, Yang et al. eLife 2014, Wang et al. Nature 2011). This issue should at least be addressed in the Discussion.*

This is an interesting and very important question that we also probed in a limited way during our studies. In C2C12 myoblasts, Wdr5 silencing resulted in stunted cell proliferation (Figure 8). When myoblasts were induced to differentiate, control myoblasts efficiently differentiated, however, Wdr5 silenced myoblasts failed to form multi-nucleated myofibers (Figure 9)and induce myogenin and MHC- two markers of differentiation we have used in this manuscript (Figure 10). More important for this question, we did not see any myogenin or MHC expression at Day 0 (in proliferative condition) when Wdr5 was silent. Thus, at least in myoblasts in our hands, Wdr5 knockdown is not sufficient to cause exit from self-renewal and differentiation. Our data show that Wdr5 is required for both the proliferative phase and differentiation process in myoblasts. This is not surprising since Wdr5 is an essential intermediate in many transcriptional events during both proliferation and differentiation states. For example, in myoblasts, Wdr5 has been shown to associate with Pax7 and regulate myoblast proliferation. During differentiation, MyoD and other transcription factors are known to recruit MLL, KAT2A-B and Set complexes of which Wdr5 is a core member to the *Myog* promoter to promote differentiation. We have included the data here, but have not included it in the manuscript per se, because it seemed a bit tangential to the logic of the already complex story. At the discretion of the editor and reviewers, we could include it in the paper.

Author response image 1.Wdr5 silencing affects proliferation rate in C2C12 myoblasts.**DOI:**
http://dx.doi.org/10.7554/eLife.17985.026

Author response image 2.Wdr5 silencing suppresses myotube formation in C2C12 myoblasts.<Figure 9>**DOI:**
http://dx.doi.org/10.7554/eLife.17985.027

Author response image 3.Wdr5 is required expression of myogenin (MyoG) and myosin (MHC) during differentiation in C2C12 myoblasts.**DOI:**
http://dx.doi.org/10.7554/eLife.17985.028

*2) Are there any effects on proliferation rate due to loss of WDR5 phosphorlyation under proliferative culture conditions?*

We do not see any effect on proliferation rate upon either PASK inhibition (Manuscript Figure 1—figure supplement 4) or Wdr5 S49A expression in myoblasts (Figure 11).

Author response image 4.C2C12 cells were infected with retrovirus expressing GFP (Control) or WT, S49A or S49E‐Wdr5 cDNAs. 24 hrs after infection, puromycin selection was performed for four days.1000 cells from each samples were plated into 96 well plates and after two days, total cell number was counted for each cell types.**DOI:**
http://dx.doi.org/10.7554/eLife.17985.029

*3) PASK knockdown had no effect on H3K4me3 signature at the MyoD promoter (or expression of MyoD) although it did influence H3K4me3 and MyoD occupancy at MyoG promoter to drive the expression of MyoG in response to differentiation cues. While MyoD is required for the driving myogenesis, is it specifically necessary for PASK dependent changes in trimethylation of H3K4 at the MyoG promoter? In other word.s given the multiple positive reinforcing feedback loops in the proposed model, could it be that MyoD acts cooperatively with WDR5 to change chromatin accessibility? The hierarchical relationship between these factors can be clarified. For example, are there differences in the observed changes in chromatin signatures with pask inhibition or deletion upon MyoD knockdown?*

This is a very interesting mechanistic question. While the interplay between MyoD and pWdr5 is an important and complex area, we have tried to address this question partially here. MyoD is extensively involved in remodeling of the *Myog* promoter at the onset of differentiation by recruiting MLL complexes, HATs and SWI/SNF complexes. MyoD serves a pioneering function at the *Myog* promoter to activate its transcription in response to differentiation signaling (Cao et al., 2006). While MyoD occupies *Myog* promoter during the proliferative phase, signaling inputs are required to stimulate chromatin remodeling and activation of MyoG transcription (Cao et al., 2006). On the other hand, inhibition of MyoD aborts the differentiation process. Thus, we hypothesize that MyoD acts synergistically with signaling inputs such as those that activate the PASK-Wdr5 pathway to remodel *Myog* promoter. We tested this by examining levels of H3K4me3 on the *Myog* promoter in control or *Myod1*- silenced C2C12 before and after the onset of differentiation. This was done in cells expressing WT, S49A and S49E Wdr5 as a surrogate for activation or inactivation of PASK signaling (Figure 12). We observed that knockdown of *Myod1* partially prevented H3K4me3 induction on the *Myog* promoter in all samples, including Wdr5_S49E_. We also examined the effect of *Myod1* knockdown on differentiation in Wdr5 WT, S49A or S49E cells. Again, we noticed that in non-targeting siRNA (Control) samples, Wdr5_S49E_ expressing cells made robust myofibers and Wdr5_S49A_ expressing cells were defective in myotube formation (Figure 13). *Myod1* knockdown, however, significantly suppressed myotube formation in Wdr5_S49E_ expressing cells. Hence, we think that Wdr5 phosphorylation is not sufficient to overcome the loss of MyoD function and that MyoD acts synergistically with phosphorylated WDR5 to induce epigenetic changes at the *Myog* promoter.

Author response image 5.H3K4me3 ChIP was performed from C2C12 myoblasts after control or MyoD knockdown expressing WT, S49A or S49E WDR5 mutants at Day 0 and Day 1 of differentiation.* P<0.05 between Wdr5_S49E_ vs. Wdr5_WT_ at Day 0 in control samples.**DOI:**
http://dx.doi.org/10.7554/eLife.17985.030

Author response image 6.Representative images of C2C12 myoblasts at Day 1 of differentiation after control or MyoD knockdown in WT, S49A or S49E WDR5 mutants expressing cells.**DOI:**
http://dx.doi.org/10.7554/eLife.17985.031